# Federated Generalized Bayesian Learning via Distributed Stein Variational Gradient Descent

## Abstract

This paper introduces Distributed Stein Variational Gradient Descent (DSVGD), a non-parametric generalized Bayesian inference framework for federated learning. DSVGD maintains a number of non-random and interacting particles at a central server to represent the current iterate of the model global posterior. The particles are iteratively downloaded and updated by one of the agents with the end goal of minimizing the global free energy. By varying the number of particles, DSVGD enables a flexible trade-off between per-iteration communication load and number of communication rounds. DSVGD is shown to compare favorably to benchmark frequentist and Bayesian federated learning strategies in terms of accuracy and scalability with respect to the number of agents, while also providing well-calibrated, and hence trustworthy, predictions.

## 1 Introduction

Federated learning refers to the collaborative training of a machine learning model across agents with distinct data sets, and it applies at different scales, from industrial data silos to mobile devices (Kairouz et al., 2019). While some common challenges exist, such as the general statistical heterogeneity – "non-iidnes" – of the distributed data sets, each setting also brings its own distinct problems. In this paper, we are specifically interested in a small-scale federated learning setting consisting of mobile or embedded devices, each having a limited data set and running a small-sized model due to their constrained memory. As an example, consider the deployment of health monitors based on data from smart-watch ECG data. In this context, we argue that it is essential to tackle the following challenges, which are largely not addressed by existing solutions:

• *Trustworthiness*: In applications such as personal health assistants, the learning agents' recommendations need to be reliable and trustworthy, e.g., to decide when to contact a doctor in case of a possible emergency;

• *Number of communication rounds*: When models are small, the payload per communication round may not be the main contributor to the overall latency of the training process. In contrast, accommodating many communication rounds requiring arbitrating channel access among multiple devices may yield slow wall-clock time convergence (Lin et al., 2020).

Most existing federated learning algorithms, such as Federated Averaging (FedAvg) (McMahan et al., 2017), are based on frequentist principles, relying on the identification of a single model parameter vector. Frequentist learning is known to be unable to capture epistemic uncertainty, yielding overconfident decisions (Guo et al., 2017). Furthermore, the focus of most existing works is on reducing the load per-communication round via compression, rather than decreasing the number of rounds by providing more informative updates at each round (Kairouz et al., 2019). This paper introduces a trustworthy solution that is able to reduce the number of communication rounds via a non-parametric variational inference-based implementation of federated Bayesian learning.

Federated Bayesian learning has the general aim of computing the global posterior distribution in the model parameter space. Existing decentralized, or federated, Bayesian learning protocols are either based on **Variational Inference (VI)** (Angelino et al., 2016; Neiswanger et al., 2015; Broderick et al., 2013; Corinzia & Buhmann, 2019b) or **Monte Carlo (MC) sampling** (Ahn et al., 2014; Mesquita et al., 2020; Wei & Conlon, 2019). State-of-the-art methods in either category include **Partitioned Variational Inference (PVI)**, which has been recently introduced as a unifying distributed VI framework that relies on the optimization over parametric posteriors; and **Distributed Stochastic Gradient Langevin Dynamics (DSGLD)**, which is an MC sampling technique that maintains a number of Markov chains updated via local Stochastic Gradient Descent (SGD) with the addition of

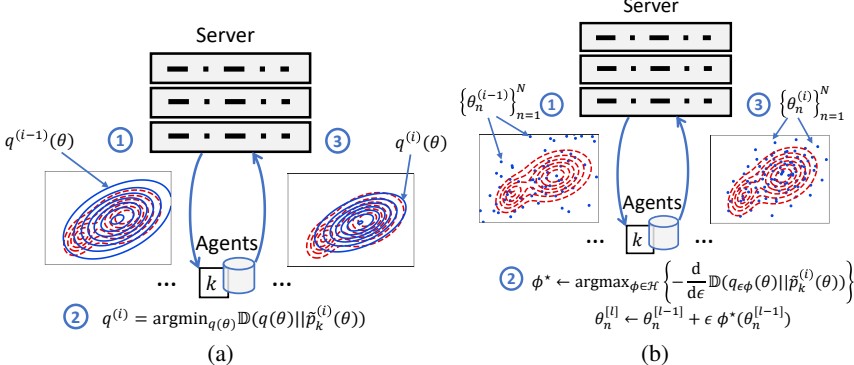

Figure 1: Federated learning across $K$ agents equipped with local datasets and assisted by a central server: (a) in DVI agents exchange the current model posterior $q^{(i)}(\theta)$ with the server, while (b) in DSVGD agents exchange particles $\{\theta_n\}_{n=1}^N$ providing a non-parametric estimate of the posterior.

Gaussian noise (Ahn et al., 2014; Welling & Teh, 2011). The performance of VI-based protocols is generally limited by the bias entailed by the variational approximation, while MC sampling is slow and suffers from the difficulty of assessing convergence (Angelino et al., 2016).

**Stein Variational Gradient Descent (SVGD)** has been introduced in (Liu & Wang, 2016) as a non-parametric Bayesian framework that approximates a target posterior distribution via non-random and interacting particles. SVGD inherits the flexibility of non-parametric Bayesian inference methods, while improving the convergence speed of MC sampling (Liu & Wang, 2016). By controlling the number of particles, SVGD can provide flexible performance in terms of bias, convergence speed, and per-iteration complexity. This paper introduces a novel non-parametric distributed learning algorithm, termed **Distributed Stein Variational Gradient Descent (DSVGD)**, that transfers the mentioned benefits of SVGD to federated learning.

As illustrated in Fig. 1, DSVGD targets a generalized Bayesian learning formulation, with arbitrary loss functions (Knoblauch et al., 2019); and maintains a number of non-random and interacting particles at a central server to represent the current iterate of the global posterior. At each iteration, the particles are downloaded and updated by one of the agents by minimizing a local free energy functional before being uploaded to the server. DSVGD is shown to enable (*i*) a trade-off between per-iteration communication load and number of communication rounds by varying the number of particles; while (*ii*) being able to make trustworthy decisions through Bayesian inference.

## 2 SYSTEM SET-UP

We consider the federated learning set-up in Fig. 1, where each agent $k = 1, \ldots, K$ has a distinct local dataset with associated training loss $L_k(\theta)$ for model parameter $\theta$. The agents communicate through a central node with the goal of computing the global posterior distribution $q(\theta)$ over the shared model parameter $\theta \in \mathbb{R}^d$ for some prior distribution $p_0(\theta)$ (Angelino et al., 2016). Specifically, following the generalized Bayesian learning framework (Knoblauch et al., 2019), the agents aim at obtaining the distribution $q(\theta)$ that minimizes the **global free energy**

$$\min_{q(\theta)} \left\{ F(q(\theta)) = \sum_{k=1}^{K} \mathbb{E}_{\theta \sim q(\theta)}[L_k(\theta)] + \alpha \mathbb{D}(q(\theta)||p_0(\theta)) \right\}, \tag{1}$$

where $\alpha > 0$ is a temperature parameter. The (generalized, or Gibbs) **global posterior** $q_{opt}(\theta)$ solving problem (1) must strike a balance between minimizing the sum loss function (first term in $F(q)$) and the model complexity defined by the divergence from a reference prior (second term in $F(q)$). It is given as

$$q_{opt}(\theta) = \frac{1}{Z} \cdot \tilde{q}_{opt}(\theta), \text{ with } \tilde{q}_{opt}(\theta) = p_0(\theta) \exp\left(-\frac{1}{\alpha} \sum_{k=1}^{K} L_k(\theta)\right), \tag{2}$$

where we denoted as $Z$ the normalization constant. It is useful to note that the global free energy can also be written as the scaled KL $F(q(\theta)) = \alpha \mathbb{D}(q(\theta)||\tilde{q}_{opt}(\theta))$.

The main challenge in computing the optimal posterior $q_{opt}(\theta)$ in a distributed manner is that each agent $k$ is only aware of its local loss $L_k(\theta)$. By exchanging information through the server, the $K$ agents wish to obtain an estimate of the global posterior (2) without disclosing their local datasets neither to the server nor to the other agents. In this paper, we introduce a novel non-parametric distributed generalized Bayesian learning framework that addresses this challenge by integrating Distributed VI (DVI) and SVGD (Liu & Wang, 2016).

## 3 DISTRIBUTED VARIATIONAL INFERENCE

In this section, we describe a general Expectation Propagation (EP)-based framework (Vehtari et al., 2020), which we term as DVI, that aims at computing the global posterior in a federated fashion (Bui et al., 2018; Corinzia & Buhmann, 2019b). DVI starts from the observation that the posterior (2) factorizes as the product

$$q(\theta) = p_0(\theta) \prod_{k=1}^{K} t_k(\theta), \qquad (3)$$

where the term $t_k(\cdot)$ is given by the scaled local likelihood $\exp(\alpha^{-1}L_k(\theta))/Z$. Since the normalization constant $Z$ depends on all data sets, the true scaled local likelihood $t_k(\cdot)$ cannot be directly computed at agent $k$. The idea of DVI is to iteratively update **approximate likelihood** factors $t_k(\theta)$ for $k = 1, ..., K$ by means of local optimization steps at the agents and communication through the server, with the aim of minimizing the global free energy (1) over distribution (3).

We give here the standard implementation of DVI in which a single agent is schedule at each time, although parallel implementations are possible and discussed below. Accordingly, at each communication round $i = 1, 2, ...$, the server maintains the current iterate $q^{(i-1)}(\theta)$ of the global posterior, and schedules an agent $k \in \{1, 2, ..., K\}$, which proceeds as follows:

**1.** Agent $k$ downloads the current global variational posterior distribution $q^{(i-1)}(\theta)$ from the server (see Fig. 1(a), step ①);

**2.** Agent $k$ updates the global posterior by minimizing the local free energy $F_k^{(i)}(q(\theta))$ (see Fig. 1(a), step ②)

$$q^{(i)}(\theta) = \underset{q(\theta)}{\text{argmin}} \left\{ F_k^{(i)}(q(\theta)) = \mathbb{E}_{\theta \sim q(\theta)}[L_k(\theta)] + \alpha \mathbb{D}(q(\theta)||\hat{p}_k^{(i)}(\theta)) \right\}, \qquad (4)$$

where we have defined the (unnormalized) **cavity distribution** $\hat{p}_k^{(i)}(\theta)$ as

$$\hat{p}_k^{(i)}(\theta) = \frac{q^{(i-1)}(\theta)}{t_k^{(i-1)}(\theta)}. \qquad (5)$$

The cavity distribution $\hat{p}_k^{(i)}(\theta)$, which removes the contribution of the current approximate likelihood of agent $k$ from the current global posterior iterate, serves as a prior for the update in (4). In a manner similar to (2), the local free energy is minimized by the **tilted distribution** $p_k^{(i)}(\theta) \propto \tilde{p}_k^{(i)}(\theta)$ with

$$\tilde{p}_k^{(i)}(\theta) = \hat{p}_k^{(i)}(\theta) \exp\left( -\frac{1}{\alpha} L_k(\theta) \right); \qquad (6)$$

**3.** Agent $k$ sends the updated posterior $q^{(i)}(\cdot) = p_k^{(i)}(\cdot)$ to the server (see Fig. 1(a), step ③), and updates its approximate likelihood accordingly as

$$t_k^{(i)}(\theta) = \frac{q^{(i)}(\theta)}{q^{(i-1)}(\theta)} t_k^{(i-1)}(\theta); \qquad (7)$$

Finally, non-scheduled agents $k' \neq k$ set $t_{k'}^{(i)}(\theta) = t_{k'}^{(i-1)}(\theta)$, and the server sets the next iterate as $q^{(i)}(\theta)$. We have the following key property of DVI.

**Theorem 1.** *The global posterior $q_{opt}(\theta)$ in (2) is the unique fixed point of the DVI algorithm.*

The fixed-point property in Theorem 1 can be verified directly by setting $q^{(i-1)}(\theta) = q_{opt}(\theta)$ and $t_k^{(i-1)}(\theta) = \exp(\alpha^{-1}L_k(\theta))/Z$ and by observing that this leads to the fixed point condition $q^{(i)}(\theta) = q^{(i-1)}(\theta) = q_{opt}(\theta)$. The proof is provided in Sec. A.6. Importantly, this property is not tied to the sequential implementation detailed above, and it applies also if multiple devices are scheduled in parallel, as long as one sets the next iterate as $q^{(i)}(\theta) = p_0(\theta) \prod_{k \in \mathcal{K}^{(i)}} t_k^{(i)}(\theta) \prod_{k' \notin \mathcal{K}^{(i)}} t_{k'}^{(i)}(\theta)$, where $\mathcal{K}^{(i)}$ denotes the set of scheduled agents at communication round $i$ and we have $t_{k'}^{(i)}(\theta) = t_{k'}^{(i-1)}(\theta)$ and $t_k^{(i)}(\theta)$ updated following (7).

## 4 PRELIMINARIES

In this section, we briefly review PVI, which serves as an important benchmark, and SVGD, on which we build the proposed Bayesian federated learning solution.

### 4.1 PARTITIONED VARIATIONAL INFERENCE

The exact minimization of the local free energy function (4) assumed by DVI is often not tractable. To address this problem, in its most typical form, PVI constrains the local free energy minimization (4) to the space of parametric distributions that factorize as $q(\theta|\eta) = p_0(\theta|\eta_0) \prod_{k=1}^{K} t_k(\theta|\eta_k)$, where prior $p_0(\cdot|\eta_0) = \mathrm{ExpFam}(\cdot|\eta_0)$ and approximate likelihood $t_k(\cdot|\eta_k) = \mathrm{ExpFam}(\cdot|\eta_k)$ are selected from the same exponential-family distribution, with natural parameters $\eta_0$ and $\eta_k$, respectively. PVI follows the same steps as DVI with the caveat that the local free energy (4) for agent $k$ is minimized over the natural parameter $\eta$. This can be done efficiently, albeit approximately, using for e.g., natural gradient descent (Amari, 1998).

The bias imposed by the parametrization in PVI significantly affects the quality of the approximation of the obtained posterior $q(\theta)$ with respect to the true global posterior $q_{opt}(\theta)$ in the presence of model misspecification. In this case, the fixed point property in Theorem 1 no longer applies.

### 4.2 STEIN VARIATIONAL GRADIENT DESCENT (SVGD)

SVGD tackles the minimization of the (scaled) free energy functional $\mathbb{D}(q(\theta)||\tilde{p}(\theta))$, for an unnormalized target distribution $\tilde{p}(\theta)$, over a non-parametric generalized posterior $q(\theta)$ defined over the model parameters $\theta \in \mathbb{R}^d$. The posterior $q(\theta)$ is represented by a set of particles $\{\theta_n\}_{n=1}^{N}$, with $\theta_n \in \mathbb{R}^d$. In practice, an approximation of $q(\theta)$ can be obtained from the particles $\{\theta_n\}_{n=1}^{N}$ through a Kernel Density Estimator (KDE) as $q(\theta) = N^{-1} \sum_{n=1}^{N} \mathrm{K}(\theta, \theta_n)$ for some kernel function $\mathrm{K}(\cdot, \cdot)$ (Bishop, 2006). The particles are iteratively updated through a series of transformations that are optimized to minimize the free energy. The transformations are restricted to lie within the unit ball of a Reproducing Kernel Hilbert Space (RKHS) $\mathcal{H}^d = \mathcal{H} \times \ldots \times \mathcal{H}$. It is shown by Liu & Wang (2016) that this optimization yields the SVGD update

$$\theta_n^{[l]} \leftarrow \theta_n^{[l-1]} + \frac{\epsilon}{N} \sum_{j=1}^{N} [\mathrm{k}(\theta_j^{[l-1]}, \theta_n^{[l-1]}) \nabla_{\theta_j} \log \tilde{p}(\theta_j^{[l-1]}) + \nabla_{\theta_j} \mathrm{k}(\theta_j^{[l-1]}, \theta_n^{[l-1]})] \tag{8}$$

for $n = 1, \ldots, N$, where $\mathrm{k}(\cdot, \cdot)$ is the positive definite kernel associated with RKHS $\mathcal{H}$. The first term in the update (8) drives the particles towards the regions of the target distribution $\tilde{p}(\theta)$ with high probability, while the second term drives the particles away from each other, encouraging exploration in the model parameter space. It is known that, in the asymptotic limit of a large number $N$ of particles, the empirical distribution encoded by the particles $\{\theta_n^{[l]}\}_{n=1}^{N}$ converges to the normalized target distribution $p(\theta) \propto \tilde{p}(\theta)$ (Liu, 2017b).

## 5 DISTRIBUTED STEIN VARIATIONAL GRADIENT DESCENT

In this section, we introduce DSVGD, a novel distributed algorithm that tackles the generalized Bayesian inference problem (1) via DVI over a non-parametric particle-based representation of the global posterior. As illustrated in Fig. 1(b), DSVGD is based on the iterative optimization of local free energy functionals (4) via SVGD (see Sec. 4), and on the exchange of particles between the central server and agents. Given the flexibility of the non-parametric form of the posterior, DSVGD doesn't suffer from the bias caused by the parametrization assumed by PVI. As a result, in the limit of a sufficiently large number of particles, DSVGD benefits from the fixed point property of DVI stated in Theorem 1, recovering the true global posterior as a fixed point of its iterations. Furthermore, as we will discuss, DSVGD enables devices to exchange more informative messages regarding the current iterate of the posterior by increasing the number of particles. This can in turn reduce the number of communication rounds and the overall communication load to convergence, at the cost of a larger per-round load. In this regard, we note that, in practice, a small number of particles is sufficient to obtain state-of-the-art performance (Liu & Wang, 2016), as verified in Sec. 7.

In order to facilitate the presentation, we first introduce a simpler version of DSVGD that has the practical drawback of requiring each agent to store a number of particles that increases linearly with the number of iterations in which the agent is scheduled. Then, we present a more practical algorithm, for which the memory requirements do not scale with the number of iterations as each agent must only memorize a set of $N$ local particles across different iterations. Algorithmic table for

---

**Algorithm 1:** Distributed Stein Variational Gradient Descent (DSVGD)

---

**Input:** prior $p_0(\theta)$, local loss functions $\{L_k(\theta)\}_{k=1}^K$, temperature $\alpha > 0$, kernels $\mathrm{K}(\cdot, \cdot)$ and $\mathrm{k}(\cdot, \cdot)$
**Output:** global approximate posterior $q(\theta) = N^{-1} \sum_{n=1}^N \mathrm{K}(\theta, \theta_n)$

---

**1** **initialize** $q^{(0)}(\theta) = p_0(\theta)$; $\{\theta_n^{(0)}\}_{n=1}^N \overset{\text{i.i.d}}{\sim} p_0(\theta)$; $\{\theta_{k,n}^{(0)} = \theta_n^{(0)}\}_{n=1}^N$ and $t_k^{(0)}(\theta) = 1$ for $k = 1, \ldots, K$
**2** **for** $i = 1, \ldots, I$ **do**
**3** $\quad$ Server schedules an Agent $k$
**4** $\quad$ Agent $k$ downloads current global particles $\{\theta_n^{(i-1)}\}_{n=1}^N$ from server
**5** $\quad$ Agent $k$ obtains updated global particles $\{\theta_n^{(i)}\}_{n=1}^N$ using (13), $\{\theta_n^{(i-1)}\}_{n=1}^N$ and $\{\theta_{k,n}^{(i-1)}\}_{n=1}^N$
**6** $\quad$ Agent $k$ sends the updated global particles $\{\theta_n^{(i)}\}_{n=1}^N$ to the server
**7** $\quad$ Agent $k$ carries distillation to obtain $\{\theta_{k,n}^{(i)}\}_{n=1}^N$ encoding $t_k^{(i)}(\theta)$ using (17) and $\{\theta_n^{(i)}\}_{n=1}^N$
**8** **end**
**9** **return** $q(\theta) = N^{-1} \sum_{n=1}^N \mathrm{K}(\theta, \theta_n^{(I)})$

---

Unconstrained-DSVGD (U-DSVGD) in addition to discussions on complexity and convergence, can be found respectively in Sec. A.1 and Sec. A.4 in the supplementary materials. A direct extension of DSVGD, termed Parallel-DSVGD (P-DSVGD), where multiple agents are scheduled per round can be found in Sec. A.5 of the Appendix.

## 5.1 U-DSVGD

In this section, we present a simplified DSVGD variant, which we refer to as U-DSVGD. We follow the standard implementation of DVI with a single agent $k$ scheduled at each communication round $i = 1, 2, \ldots$, although, as discussed, parallel implementations are also possible. Let us define as $\mathcal{I}_k^{(i)} \subseteq \{1, \ldots, i\}$ the subset of rounds at which agent $k$ is scheduled prior, and including, iteration $i$. At the beginning of each round $i$, the server maintains the iterate of the current global particles $\{\theta_n^{(i-1)}\}_{n=1}^N$, while each agent $k$ keeps a local buffer of particles $\{\theta_n^{(j-1)}, \theta_n^{(j)}\}_{n=1}^N$ for all previous rounds $j \in \mathcal{I}_k^{(i-1)}$ at which agent $k$ was scheduled. The growing memory requirements at the agents will be dealt with by the final version of DSVGD to be introduced in Sec. 5.2. Furthermore, as illustrated in Fig. 1(b), at each iteration $i$, U-DSVGD schedules an agent $k \in \{1, 2, \ldots, K\}$ and carries out the following steps.

**1.** Agent $k$ downloads the current global particles $\{\theta_n^{(i-1)}\}_{n=1}^N$ from the server (see Fig. 1(b), step ①)) and includes them in the local buffer.

**2.** Agent $k$ updates each downloaded particle as

$$\theta_n^{[l]} \leftarrow \theta_n^{[l-1]} + \epsilon \phi(\theta_n^{[l-1]}), \text{ for } l = 1, \ldots, L, \tag{9}$$

where $L$ is the number of local iterations; $[l]$ denotes the *local iteration* index; we have the initialization $\theta_n^{[0]} = \theta_n^{(i-1)}$; and the function $\phi(\cdot)$ is to be optimized within the unit ball of a RKHS $\mathcal{H}^d$. The function $\phi(\cdot)$ is specifically optimized to maximize the steepest descent decrease of a particle-based approximation of the local energy (4). To elaborate, we denote as $q^{(i-1)}(\theta) = \sum_{n=1}^N \mathrm{K}(\theta, \theta_n^{(i-1)})$ the KDE of the current global posterior iterate encoded by particles $\{\theta_n^{(i-1)}\}_{n=1}^N$. Adopting the factorization (3) for the global posterior (cf. (7)), we define the current local approximate likelihood

$$t_k^{(i-1)}(\theta) = \prod_{j \in \mathcal{I}_k^{(i-1)}} \frac{q^{(j)}(\theta)}{q^{(j-1)}(\theta)} = \frac{q^{(i-1)}(\theta)}{q^{(i-2)}(\theta)} t_k^{(i-2)}(\theta). \tag{10}$$

Note that (10) can be computed using *all* the particles in the buffer at agent $k$ at iteration $i$. Finally, the (unnormalized) tilted distribution $\tilde{p}_k^{(i)}$ (cf. (6)) is written as

$$\tilde{p}_k^{(i)}(\theta) = \frac{q^{(i-1)}(\theta)}{t_k^{(i-1)}(\theta)} \exp\left(-\frac{1}{\alpha} L_k(\theta)\right). \tag{11}$$

Following SVGD, the update (9) is optimized to maximize the steepest descent decrease of the Kullback–Leibler (KL) divergence between the approximate global posterior $q_{\epsilon\phi}^{[l]}(\theta)$ encoded via particles $\{\theta_n^{[l]}\}_{n=1}^N$ and the tilted distribution $\tilde{p}_k^{(i)}(\theta)$ in (11) (see Fig. 1(b), step ②)), i.e.,



Figure 2: Gaussian toy example with uniform prior and $K = 2$. Dashed lines represent local posteriors, the shaded area represents the true global posterior, while the solid blue line is the approximate posterior obtained using a KDE over the particles. DSVGD schedules agent 1 and 2 at odd and even number of communication rounds $i$, respectively.

$$\phi^\star(\cdot) \leftarrow \arg\max_{\phi(\cdot) \in \mathcal{H}^d} \left\{ -\frac{d}{d\epsilon} \mathbb{D}(q_{\epsilon\phi}^{[l-1]}(\theta) || \tilde{p}_k^{(i)}(\theta)), \quad \text{s.t.} \quad ||\phi||_{\mathcal{H}^d} \leq 1 \right\}. \tag{12}$$

Thus, recalling (8), the particles are updated as

$$\theta_n^{[l]} \leftarrow \theta_n^{[l-1]} + \frac{\epsilon}{N} \sum_{j=1}^{N} [\mathrm{k}(\theta_j^{[l-1]}, \theta_n^{[l-1]}) \nabla_{\theta_j} \log \tilde{p}_k^{(i)}(\theta_j^{[l-1]}) + \nabla_{\theta_j} \mathrm{k}(\theta_j^{[l-1]}, \theta_n^{[l-1]})], \text{for } l=1,\dots,L. \tag{13}$$

**3.** Agent $k$ sets $\theta_n^{(i)} = \theta_n^{[L]}$ for $n = 1, \dots, N$. Particles $\{\theta_n^{(i)}\}_{n=1}^N$ are added to the buffer and sent to the server (see Fig. 1(b), step ③) that updates the current global particles as $\{\theta_n\}_{n=1}^N = \{\theta_n^{(i)}\}_{n=1}^N$.

In order to implement the described U-DSVGD algorithm, we need to compute the gradient in (13) at agent $k$. First, by (11), we have

$$\nabla_\theta \log \tilde{p}_k^{(i)}(\theta) = \nabla_\theta \log q^{(i-1)}(\theta) - \nabla_\theta \log t_k^{(i-1)}(\theta) - \frac{1}{\alpha} \nabla_\theta L_k(\theta). \tag{14}$$

Using (10), the second gradient term can be obtained in a recursive manner using the local buffer as

$$\nabla_\theta \log t_k^{(i-1)}(\theta) = \begin{cases} \nabla_\theta \log t_k^{(i-2)}(\theta) \text{ if agent } k \text{ not scheduled at iteration } (i-1) \\ \nabla_\theta \log t_k^{(i-2)}(\theta) + \nabla_\theta \log q^{(i-1)}(\theta) - \nabla_\theta \log q^{(i-2)}(\theta) \text{ otherwise.} \end{cases} \tag{15}$$

Finally, the gradients $\nabla_\theta \log q^{(j)}(\theta)$ can be directly computed from the KDE expression of $q^{(j)}(\theta)$, with initializations $t^{(0)}(\theta) = 1$ and $q^{(0)}(\theta) = p_0(\theta)$. The inner loop of U-DSVGD inherits the asymptotic convergence properties of SVGD in terms of local free energies, but existing results do not imply that the global free energy decreases across the iterations. This result is provided in the next theorem, whose precise formulation can be found in Sec. A.6 of the Appendix.

**Theorem 2** (Guaranteed per-iteration decrease of the global free energy.)**.** *The decrease in the global free energy from local iteration $l$ to $l+1$ during communication round $i$ for which agent $k$ is scheduled can be lower bounded as*

$$F(q^{[l]}(\theta)) - F(q^{[l+1]}(\theta)) \geq \alpha\epsilon S(q^{[l]}, p_k^{(i)})(1 - \epsilon\gamma) - 2\alpha(K-1)l_{\max}^{(i)} \sqrt{2\mathbb{D}(q^{[l+1]}||q^{[l]})}, \tag{16}$$

*where $l_{\max}^{(i)} = \sup_\theta \max_{m \neq k} |\log(t_m^{(i-1)}(\theta)) \cdot \exp(\frac{1}{\alpha} L_m(\theta))|$, $S(q,p)$ denotes the Kernelized Stein Discrepancy between distributions $q$ and $p$ (Liu et al., 2016), and $\gamma$ is a constant depending on the RKHS kernel and the target distribution.*

The first term in bound (16) quantifies the decrease in the local free energy at agent $k$, which depends on the "distance" between current iterate $q^{[l]}$ and the local target given by the tilted distribution $p_k^{(i)}(\theta)$; while the second term quantifies the effect of the update on the local free energies of other agents. In the presence of only one agent, the second terms reduce to zero, and one recovers the upper bound on the guaranteed per-iteration improvement for SVGD derived in Korba et al. (2020).

### 5.2 DSVGD

In this section, we describe the final version of DSVGD, which, unlike U-DSVGD, requires each agent $k$ to maintain only $N$ *local* particles $\{\theta_{k,n}^{(i)}\}_{n=1}^N$ across the communication rounds $i = 1, 2, \dots$. To this end, in each round $i$, at the end of the $L$ local SVGD updates in (13), DSVGD carries out a

Figure 4: Accuracy for Bayesian logistic regression with (left) $K = 2$ agents and (right) $K = 20$ agents as function of the number of communication rounds $i$ ($N = 6$ particles, $L = L' = 200$).

form of model **distillation** (Hinton et al., 2015; Chen & Chao, 2020) via SVGD. Specifically, $L'$ additional SVGD steps are used to approximate the term $t_k^{(i)}(\theta)$ using the $N$ local particles $\{\theta_{k,n}^{(i)}\}_{n=1}^N$. It is noted that this approximation step is not necessarily harmful to the overall performance, since describing the factor $t_k^{(i)}(\theta)$ with fewer particles can have a denoising effect acting as a regularizer.

DSVGD operates as U-DSVGD apart from the computation of the gradient in (14) and the management of the local particle buffers. The key idea is that, instead of using the recursion (15) to compute (14), DSVGD computes the gradient $\nabla_\theta \log t_k^{(i-1)}(\theta)$ from the KDE $t_k^{(i-1)}(\theta) = \sum_{n=1}^N \mathrm{K}(\theta, \theta_{k,n}^{(i-1)})$ based on the local particles $\{\theta_{k,n}^{(i-1)}\}_{n=1}^N$ in the buffer. At the end of each round $i$, the local particles $\{\theta_{k,n}^{(i-1)}\}_{n=1}^N$ are updated by running $L'$ local SVGD iterations with target given by the updated local factor $t_k^{(i)}(\theta) = \frac{q^{(i)}(\theta)}{q^{(i-1)}(\theta)} t_k^{(i-1)}(\theta)$. This amounts to the updates

$$\theta_{k,n}^{[l']} \leftarrow \theta_{k,n}^{[l'-1]} + \frac{\epsilon'}{N} \sum_{j=1}^N [k(\theta_{k,j}^{[l'-1]}, \theta_{k,n}^{[l'-1]}) \nabla_{\theta_j} \log t_k^{(i)}(\theta) + \nabla_{\theta_j} k(\theta_{k,j}^{[l'-1]}, \theta_{k,n}^{[l'-1]})], \qquad (17)$$

for $l' = 1, \dots, L'$ and some learning rate $\epsilon'$, where the gradient $\nabla_\theta \log t_k^{(i)}(\theta) = \nabla_\theta \log q^{(i)}(\theta) + \nabla_\theta \log t_k^{(i-1)}(\theta) - \nabla_\theta \log q^{(i-1)}(\theta)$ can be directly computed using KDE based on the available particles $\{\theta_n^{(i)}\}_{n=1}^N$ (updated global particles), $\{\theta_{k,n}^{(i-1)}\}_{n=1}^N$ (local particles) and $\{\theta_n^{(i-1)}\}_{n=1}^N$ (downloaded global particles). Finally, we note that the distillation operation can be performed after sending the updated global particles to the server and thus enabling pipelining of the $L'$ local iterations with operations at the server and other agents. DSVGD is summarized in Algorithm 1.

## 6 RELATED WORK

**Extensions of SVGD.** Since its introduction, SVGD has been extended in various directions. Most related to this work is Zhuo et al. (2018), which introduces a message-passing SVGD solution for high-dimensional latent parameter spaces by leveraging conditional independence properties in the variational posterior; and Yoon et al. (2018), which uses SVGD as the per-task base learner in a meta-learning algorithm approximating Expectation Maximization.

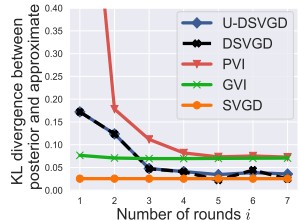

Figure 3: KL divergence between exact and approximate global posteriors as function of the number of rounds $i$ ($L = L' = 200$).

**Generalized Bayesian Inference.** Owing to its reliance on point estimates in the model parameter space, frequentist learning methods, such as Federated Stochastic Gradient Descent (FedSGD), FedAvg and their extensions (Zhang et al., 2020; Li et al., 2018; Pathak & Wainwright, 2020; Nguyen et al., 2020; Wang et al., 2020) are limited in their capacity to combat overfitting and quantify uncertainty (Guo et al., 2017; Mitros & Mac Namee, 2019; Neal, 2012; Jospin et al., 2020; MacKay, 2002). This contrasts with the generalized Bayesian inference framework that produces distributional, rather than point, estimates by optimizing the free energy functional, which is a theoretically principled bound on the generalization performance (Zhang, 2006; Knoblauch et al., 2019). Practical algorithms for generalized Bayesian inference can leverage computationally efficient scalable solutions based on either MC sampling or VI methods (Angelino et al., 2016; Alquier et al., 2016).

**Distributed MC Sampling.** The design of algorithms for distributed Bayesian learning has been so far mostly focused on one-shot, or "embarrassingly parallel", solutions under ideal communications (Jordan et al., 2019). These implement distributed MC "consensus" protocols, whereby samples from the global posterior are approximately synthesized by combining particles from local posteriors (Scott et al., 2016; Liu & Ihler, 2014). Iterative extensions, such as Weierstrass sampling (Wang &

Dunson, 2013; Rendell et al., 2018), impose consistency constraints across devices and iterations in a way similar to the Alternating Direction Method of Multipliers (ADMM) (Angelino et al., 2016). State-of-the-art results have been obtained via DSGLD (Ahn et al., 2014).

**Distributed VI Learning.** Considering first one-shot model fusion of local models, Bayesian methods have been used to deal with parameter invariance and weight matching (Yurochkin et al., 2019; Claici et al., 2020). Iterative VI such as streaming variational Bias (SVB) (Broderick et al., 2013) provide a VI-based framework for the exponential family to combine local models into global ones. PVI provides a general framework that can implement SVB, as well as online VI (Bui et al., 2018) and has been extended to multi-task learning in Corinzia & Buhmann (2019a).

## 7 EXPERIMENTS

As in Liu & Wang (2016), for all our experiments with SVGD and DSVGD, we use the Radial Basis Function (RBF) kernel $k(x, x_0) = \exp(-||x - x_0||_2^2/h)$. The bandwidth $h$ is adapted to the set of particles used in each update by setting $h = \text{med}^2/\log n$, where med is the median of the pairwise distances between the particles in the current iterate. The Gaussian kernel $K(\cdot, \cdot)$ used for the KDEs has a bandwidth equal to $0.55$. Unless specified otherwise, we use AdaGrad with momentum to choose the learning rates $\epsilon$ and $\epsilon'$ for (U-)DSVGD. Throughout, we fix the temperature parameter $\alpha = 1$ in (1). Finally, to ensure a fair comparison with distributed schemes, we run centralized schemes for the same total number $I \times L$ of iterations across all experiments. Additional results for all experiments can be found in Appendix B in the supplementary materials, which include also additional implementation details.

**Gaussian 1D mixture toy example.** We start by considering a simple one-dimensional mixture model in which the local unnormalized local posteriors $p_k(\theta) = p_0(\theta)\exp(-\alpha^{-1}L_k(\theta))$ at each agent $k$ are defined as $p_1(\theta) = p_0(\theta)\mathcal{N}(\theta|1, 4)$ and $p_2(\theta) = p_0(\theta)(\mathcal{N}(\theta|-3, 1) + \mathcal{N}(\theta|3, 2))$ and the prior $p_0(\theta)$ is uniform over $[-6, 6]$, i.e., $p_0(\theta) = \mathcal{U}(\theta|-6, 6)$. The local posteriors are shown in Fig. 2 as dashed lines, along with the global posterior $q_{opt}(\theta) \propto \tilde{q}_{opt}(\theta)$ in (2), which is represented as a shaded area. We fix the number of particles to $N = 200$. The approximate posteriors obtained from the KDE over the global particles are plotted in Fig. 2 as solid lines. It can be observed that at each round, the global posterior updated by DSVGD integrates the local likelihood

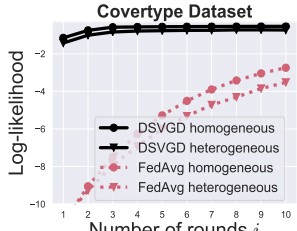

Figure 5: Log-likelihood for Bayesian logistic regression with non-iid data distributions ($N = 6, L = L' = 200$).

of the scheduled agent, while still preserving information about the likelihood of the other agent from prior iterates, until (approximate) convergence to the true global posterior $q_{opt}$, which is a normalized version of $\tilde{q}_{opt}$ in (2), is reached. Finally, in Fig. 3, we plot the KL divergence between $q(\theta)$ and $q_{opt}(\theta)$ as a function of the number of rounds. Both U-DSVGD and DSVGD exhibit similar behaviour, converging to SVGD and outperforming the parametric counterparts PVI and Global Variational Inference (GVI) (Bui et al., 2018).

**Bayesian logistic regression.** We now consider Bayesian logistic regression for binary classification using the same setting as in Gershman et al. (2012). The model parameters $\theta = [\mathbf{w}, \log(\xi)]$ include the regression weights $\mathbf{w} \in \mathbb{R}^d$ along with the logarithm of a precision parameter $\xi$. The prior is given as $p_0(\mathbf{w}, \xi) = p_0(\mathbf{w}|\xi)p_0(\xi)$, with $p_0(\mathbf{w}|\xi) = \mathcal{N}(\mathbf{w}|\mathbf{0}, \xi^{-1}\mathbf{I}_d)$ and $p_0(\xi) = \text{Gamma}(\xi|a, b)$ with $a = 1$ and $b = 0.01$. The local training loss $L_k(\theta)$ at each agent $k$ is given as $L_k(\theta) = \sum_{(\mathbf{x}_k, y_k) \in D_k} l(\mathbf{x}_k, y_k, \mathbf{w})$, where $D_k$ is the dataset at agent $k$ with covariates $\mathbf{x}_k \in \mathbb{R}^d$ and label $y_k \in \{-1, 1\}$, and the loss function $l(\mathbf{x}_k, y_k, \mathbf{w})$ is the cross-entropy. Point decisions are taken based on the maximum of the average predictive distribution. We consider the datasets Covertype and Twonorm (Gershman et al., 2012). We randomly split the training dataset into partitions of equal size among the $K$ agents. We also include FedAvg, Stochastic Gradient Langevin Dynamics (SGLD) and DSGLD for comparison. We note that FedAvg is implemented here for consistency with the other schemes by scheduling a single agent at each step. In Fig. 4, we study how the accuracy evolves as function of the number of communication rounds $i$, or number of communication rounds, across different datasets, using $N = 2$ and $N = 6$ particles. We observe that DSVGD consistently outperforms the mentioned decentralized benchmarks and that, in contrast to FedAvg and DSGLD, its performance scales well with the number $K$ of agents. Furthermore, the number $N$ of particles is seen to control the trade-off between the communication load, which increases with $N$, and the convergence speed, which improves as $N$ grows larger. Through reduction of the number

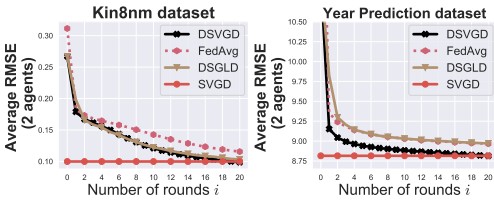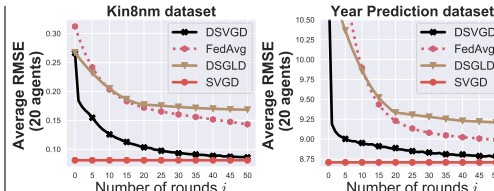

Figure 6: Average RMSE as a function of the number of communication rounds $i$ for regression using BNN with a single hidden layer of ReLUs with (left) $K = 2$ agents and (right) $K = 20$ agents ($N = 20$, $L = L' = 200$, 100 hidden neurons for the Year Prediction and 50 for Kin8nm).

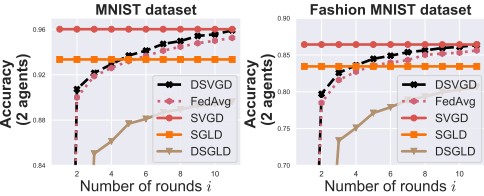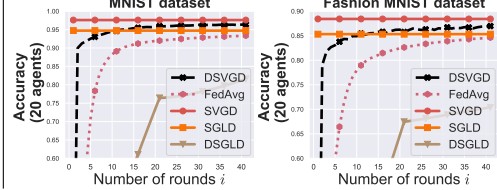

Figure 7: Multi-label classification accuracy using BNN with a single hidden layer of 100 neurons as function of $i$, or number of communication rounds, using MNIST and Fashion MNIST with (left) $K = 2$ agents and (right) $K = 20$ agents ($N = 20$, $L = L' = 200$).

of communication rouds, DSVGD can also reduce the overall communication load. For example, in the third plot in Fig. 4, DSVGD reaches an accuracy of 70% after 5 communication rounds with $N = 6$, requiring the exchange of 30 particles. In contrast, FedAvg requires around 100 rounds to obtain the same accuracy, making the total communication load much higher than that of DSVGD.

To capture heterogeneous datasets with non i.i.d. data, we now consider for different dataset partitions across $K = 4$ agents. In the homogeneous case, labels are split equally among agents, while, in the heterogeneous case, each agent stores 40% of one label and 10% of the other. DSVGD is seen in Fig. 5 to have a robust performance against heterogeneity as compared to FedAvg, whose convergence speed is severely affected. This result hinges on the fact that Bayesian learning provides a predictive distribution that is a more accurate estimate of the ground-truth posterior distribution. This is true irrespective of the level of "non-iidness": Bayesian learning can account in a principled away for all competing "explanations" provided by different devices. This is in contrast to FedAvg, whose reliance on a point estimate of the parameters yields an overconfident predictive distribution that cannot properly account for the diversity of predictions provided by different devices.

**Bayesian Neural Networks.** We now consider regression and multi-label classification with Bayesian Neural Networks (BNN) models. The experimental setup is the same as in Hernández-Lobato & Adams (2015), with the only exception that the prior of the weights is set to $p_0(\mathbf{w}) = \mathcal{N}(\mathbf{w}|0, \lambda^{-1}\mathbf{I}_d)$ with a fixed precision $\lambda = e$. We plot the average Root Mean Square Error (RMSE) for $K = 2$ and $K = 20$ agents in Fig. 6 for regres-

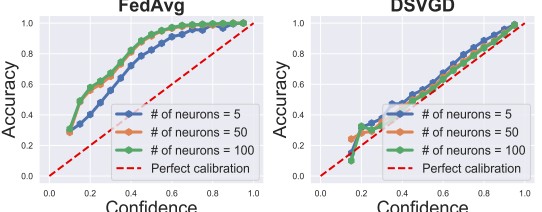

Figure 8: Reliability plots for classification using BNN with variable number of hidden neurons using fashion MNIST ($N = 20$, $I = 10$, $L = L' = 200$, $K = 20$).

sion over the Kin8nm and Year datasets, and accuracy for multi-label classification on the MNIST and Fashion MNIST datasets in Fig. 7. Confirming the results for logistic regression, DSVGD consistently outperforms the other decentralized benchmarks in terms of RMSE and accuracy, while being more robust in terms of convergence speed to an increase in the number of agents.

**Calibration.** Reliability plots are a common visual tool used to quantify and visualize model calibration (Guo et al., 2017). They report the average sample accuracy as function of the confidence level of the model. Perfect calibration yields an accuracy equal to the corresponding confidence (dashed line in Fig. 8). Fig. 8 shows the reliability plots for FedAvg and DSVGD on the Fashion MNIST dataset for the BNN setting. While increasing the number of hidden neurons negatively affects FedAvg due to overfitting, DSVGD enjoys excellent calibration even for large models and is hence able to make trustworthy predictions.

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

## A   COMPLEMENTARY MATERIALS

### A.1   ALGORITHMIC TABLES

---

**Algorithm 2:** Partitioned Variational Inference (PVI) (Bui et al., 2018)

---

**Input:** prior $p_0(\theta)$, local loss function $\{L_k(\theta)\}_{k=1}^K$, temperature $\alpha > 0$
**Output:** global posterior $q(\theta|\eta)$

---

1 **initialize** $t_k^{(0)}(\theta) = 1$ for $k = 1, \ldots, K$; $q^{(0)}(\theta) = p_0(\theta)$
2 **for** $i = 1, \ldots, I$ **do**
3      At scheduled agent $k$, download current global parameters $\eta^{(i-1)}$ from server
4      Agent $k$ solves local free energy problem in (4) to obtain new global parameters $\eta^{(i)}$
5      Agent $k$ sends $\eta^{(i)}$ to the server and **server sets** $\eta \leftarrow \eta^{(i)}$
6      Agent $k$ updates new approximate likelihood: $t_k(\theta|\eta_k^{(i)}) = \frac{q(\theta|\eta^{(i)})}{q(\theta|\eta^{(i-1)})} t_k(\theta|\eta_k^{(i-1)})$
7 **end**
8 **return** $q(\theta) = q(\theta|\eta^{(I)})$

---

---

**Algorithm 3:** Stein Variational Gradient Descent (SVGD) (Liu & Wang, 2016)

---

**Input:** target distribution $\tilde{p}(\theta)$, initial particles $\{\theta_n^{(0)}\}_{n=1}^N \sim p_0(\theta)$, kernel $k(\cdot, \cdot)$, learning rate $\epsilon$
**Output:** particles $\{\theta_n\}_{n=1}^N$ that approximates the target normalized distribution

---

1 **for** $i = 1, \ldots, L$ **do**
2      **for** $n = 1, \ldots, N$ **do**
3          $\theta_n^{(i)} \leftarrow \theta_n^{(i-1)} + \frac{\epsilon}{N} \sum_{j=1}^N [k(\theta_j^{(i-1)}, \theta_n^{(i-1)}) \nabla_{\theta_j} \log \tilde{p}(\theta_j^{(i-1)}) + \nabla_{\theta_j} k(\theta_j^{(i-1)}, \theta_n^{(i-1)})].$
4      **end**
5 **end**
6 **return** $q(\theta) = \sum_{n=1}^N K(\theta, \theta_n^{(L)})$

---

---

**Algorithm 4:** Unconstrained-Distributed Stein Variational Gradient Descent (U-DSVGD)

---

**Input:** prior $p_0(\theta)$, local loss function $\{L_k(\theta)\}_{k=1}^K$, temperature $\alpha > 0$, learning rate $\epsilon > 0$, kernels $K(\cdot, \cdot)$ and $k(\cdot, \cdot)$
**Output:** global posterior $q(\theta) = \sum_{n=1}^N K(\theta, \theta_n)$

---

1 **initialize** $t_k^{(0)}(\theta) = 1$ for $k = 1, \ldots, K$; $q^{(0)}(\theta) = p_0(\theta)$; $\{\theta_n^{(0)}\}_{n=1}^N \overset{i.i.d}{\sim} p_0(\theta)$
2 **for** $i = 1, \ldots, I$ **do**
     // New communication round:  server schedules an agent $k$
3      At scheduled agent $k$, download and memorize in local buffer current global particles $\{\theta_n^{(i-1)}\}_{n=1}^N$
4      Agent $k$ sets $\{\theta_n^{[0]} = \theta_n^{(i-1)}\}_{n=1}^N$
5      **for** $l = 1, \ldots, L$ **do**
         // Local iterations:  agent $k$ minimizes local free energy
6          Compute $\nabla_{\theta_n}^{[l]} = \nabla_\theta \log q^{(i-1)}(\theta_n^{[l-1]}) - \nabla_\theta \log t_k^{(i-1)}(\theta_n^{[l-1]}) - \frac{1}{\alpha} \nabla_\theta L_k(\theta_n^{[l-1]})$ with KDE
         $q^{(i-1)}(\theta) = \sum_{n=1}^N K(\theta, \theta_n^{(i-1)})$ and $\nabla_\theta \log t_k^{(i-1)}(\theta)$ computed using (15)
7          **for** particle $n = 1, ..., N$ **do**
8              $\Delta\theta_n \leftarrow \frac{1}{N} \sum_{j=1}^N \left[ k(\theta_j^{[l-1]}, \theta_n^{[l-1]}) \nabla_{\theta_j}^{[l]} + \nabla_{\theta_j} k(\theta_j^{[l-1]}, \theta_n^{[l-1]}) \right]$
9              $\theta_n^{[l]} \leftarrow \theta_n^{[l-1]} + \epsilon \Delta\theta_n$
10          **end**
11      **end**
12      Agent $k$ sets updated global particles $\{\theta_n^{(i)} = \theta_n^{[L]}\}_{n=1}^N$ and memorize them in the local buffer
13      Agent $k$ sends particles $\{\theta_n^{(i)}\}_{n=1}^N$ to the server and **server sets** $\{\theta_n = \theta_n^{(i)}\}_{n=1}^N$
14 **end**
15 **return** $q(\theta) = \sum_{n=1}^N K(\theta, \theta_n^{(I)})$

---

---

**Algorithm 5:** Parallel-Distributed Stein Variational Gradient Descent (P-DSVGD)

---

**Input:** prior $p_0(\theta)$, local loss functions $\{L_k(\theta)\}_{k=1}^K$, temperature $\alpha > 0$, kernels $\mathrm{K}(\cdot, \cdot)$ and $\mathrm{k}(\cdot, \cdot)$
**Output:** global approximate posterior $q(\theta) = N^{-1} \sum_{n=1}^N \mathrm{K}(\theta, \phi_n)$

---

1   **initialize** $q^{(0)}(\theta) = p_0(\theta)$; $\{\theta_n^{(0)}\}_{n=1}^N \overset{\text{i.i.d}}{\sim} p_0(\theta)$; $\{\phi_n^{(0)} = \theta_{k,n}^{(0)} = \theta_n^{(0)}\}_{n=1}^N$ and $t_k^{(0)}(\theta) = 1$ for $k = 1, \ldots, K$

2   **for** $i = 1, \ldots, I$ **do**

3     Server schedules a set $\mathcal{K}^{(i)}$ of agents in parallel

4     Agents downloads current server particles $\{\phi_n^{(i-1)}\}_{n=1}^N$ from server

5     Agents obtains updated global particles $\{\theta_n^{(i)}\}_{n=1}^N$ using (13), $\{\theta_n^{(i-1)} = \phi_n^{(i-1)}\}_{n=1}^N$ and $\{\theta_{k,n}^{(i-1)}\}_{n=1}^N$

6     Agents carries distillation to obtain $\{\theta_{k,n}^{(i)}\}_{n=1}^N$ encoding $t_k^{(i)}(\theta)$ using (17) and $\{\theta_n^{(i)}\}_{n=1}^N$

7     Agents sends the obtained local particles $\{\theta_{k,n}^{(i)}\}_{n=1}^N$ for $k \in \mathcal{K}^{(i)}$ to the server

8     Server obtains $\{\phi_n^{(i)}\}_{n=1}^N$ using (29), $\{\phi_n^{(i-1)}\}_{n=1}^N$ and $\{\theta_{k,n}^{(i)}\}_{n=1}^N$ for $k \in \mathcal{K}^{(i)}$

9   **end**

10   **return** $q(\theta) = N^{-1} \sum_{n=1}^N \mathrm{K}(\theta, \phi_n^{(I)})$

---

## A.2 A RELATIONSHIP BETWEEN PVI AND U-DSVGD

We show here that PVI with a Gaussian variational posterior $q(\theta|\eta) = \mathcal{N}(\theta|\lambda^2\eta, \lambda^2\mathbf{I}_d)$ of fixed covariance $\lambda^2\mathbf{I}_d$ and mean $\lambda^2\eta$ parametrized by natural parameter $\eta$ can be recovered as a special case of U-DSVGD. To elaborate, consider U-DSVGD with one particle $\theta_1$ (i.e., $N = 1$), an RKHS kernel that satisfies $\nabla_\theta \mathrm{k}(\theta, \theta) = 0$ and $\mathrm{k}(\theta, \theta) = 1$ (the RBF kernel is an example of such kernel) and an isotropic Gaussian kernel $K(\theta, \theta_1^{(i)}) = \mathcal{N}(\theta|\theta_1^{(i)}, \lambda^2\mathbf{I}_d)$ of bandwidth $\lambda$ used for computing the KDE of the global posterior using the particles. The U-DSVGD particles update in (13) reduces to the following single particle update:

$$\theta_1^{[l]} \leftarrow \theta_1^{[l-1]} + \epsilon \nabla_\theta \log \tilde{p}_k^{(i)}(\theta_1^{[l-1]}), \text{ for } l = 1, \dots, L, \tag{18}$$

with tilted distribution

$$\tilde{p}_k^{(i)}(\theta) \propto \frac{q^{(i-1)}(\theta)}{t_k^{(i-1)}(\theta)} \exp\left(-\frac{1}{\alpha}L_k(\theta)\right). \tag{19}$$

The numerator in (19) can be rewritten as $q^{(i-1)}(\theta) = \mathrm{K}(\theta, \theta_1^{(i-1)}) = q(\theta|\eta^{(i-1)})$ with $\eta^{(i-1)} = \lambda^{-2}\theta_1^{(i-1)}$, while the denominator can be rewritten as

$$t_k^{(i-1)}(\theta) = \prod_{j \in \mathcal{I}_k^{(i-1)}} \frac{q(\theta|\eta^{(j)})}{q(\theta|\eta^{(j-1)})} = t_k(\theta|\eta_k^{(i-1)}), \tag{20}$$

with $\eta_k^{(i-1)} = \sum_{j \in \mathcal{I}_k^{(i-1)}} \eta^{(j)} - \eta^{(j-1)}$. This recovers the PVI update (6).

## A.3 RELIABILITY PLOTS

In this part we give some background on reliability plots and Maximum Calibration Error (MCE). Reliability plots are a visual tool to evaluate model calibration (DeGroot & Fienberg, 1983; Niculescu-Mizil & Caruana, 2005). Consider a model that outputs a prediction $\hat{y}(x_i)$ and a probability $\hat{p}(x_i)$ of correct detection for an input $x_i$ with true label $y_i$. We divide the test samples into bins $\{\mathcal{B}_j\}_{j=1}^B$, each bin $\mathcal{B}_j$ containing all indices of samples whose prediction confidence falls into the interval $(\frac{j-1}{B}, \frac{j}{B}]$ where $B$ is the total number of bins. Reliability plots evaluate the accuracy as function of the confidence which are defined respectively as

$$\mathrm{acc}(\mathcal{B}_j) = \frac{1}{|\mathcal{B}_j|} \sum_{i \in \mathcal{B}_j} \mathbf{1}_{\{\hat{y}(x_i)=y_i\}}$$

$$\text{and } \mathrm{conf}(\mathcal{B}_j) = \frac{1}{|\mathcal{B}_j|} \sum_{i \in \mathcal{B}_j} \hat{p}(x_i).$$

Perfect calibration means that the accuracy is equal to the confidence across all bins. For example, given 100 predictions, each with confidence approximately 0.7, one should expect that around 70% of these predictions be correctly classified.

To compute $\hat{p}(x)$, we need the predictive probability $p(y_t|\mathbf{x}_t)$ for all samples $t \in [1; T]$. This can be obtained by marginalizing the data likelihood with respect to the weights vector $\mathbf{w}$. This marginalization is generally intractable but can be approximated for both Bayesian logistic regression and Bayesian Neural Networks as detailed in Sec. A.3.1 and Sec. A.3.2.

While reliability plots are a useful tool to visually represent the calibration of a model, it is often desirable to have a single scalar measure of miscalibration. In this paper, we use the MCE that measures the worst case deviation of the model calibration from perfect calibration (Guo et al., 2017). Mathematically, the MCE is defined as

$$\mathrm{MCE} = \max_{j \in \{1, \dots, B\}} |\mathrm{acc}(\mathcal{B}_j) - \mathrm{conf}(\mathcal{B}_j)|. \tag{21}$$

Additional numerical results using both reliability plots and MCE can be found in Sec. B.5.

### A.3.1 PREDICTIVE DISTRIBUTION FOR BAYESIAN LOGISTIC REGRESSION WITH SVGD AND DSVGD

In this section, we show how the predictive distribution for the Bayesian logistic regression experiment can be obtained when using DSVGD or SVGD. The predictive distribution provides the confidence values to be used in the calibration experiment. Given a KDE of the posterior $q(\mathbf{w}) = \sum_{n=1}^{N} \mathrm{k}(\mathbf{w}, \mathbf{w}_n)$ with $N$ particles $\{\mathbf{w}_n\}_{n=1}^{N}$ the predictive probability for Bayesian logistic regression can be estimated as

$$p(y_t = 1|\mathbf{x}_t) \approx \int p(y_t = 1|\mathbf{x}_t, \mathbf{w})q(\mathbf{w})\mathrm{d}\mathbf{w} = \sum_{n=1}^{N} \frac{1}{N(2\lambda^2\pi)^{d/2}} \int \frac{\exp(\frac{-1}{2\lambda^2}||\mathbf{w}_n - \mathbf{w}||^2)}{1 + \exp(-\mathbf{w}\mathbf{x}_t^T)}\mathrm{d}\mathbf{w}.$$

(22)

A good approximation of (22) can be obtained by replacing the logistic sigmoid function with the probit function (Bishop, 2006, Sec. 4.5), yielding

$$p(y_t = 1|\mathbf{x}_t) \approx \sum_{n=1}^{N} \frac{1}{N} \frac{1}{1 + \exp(-\kappa(\sigma^2)\mu_n)},$$

(23)

where

$$\mu_n = \mathbf{w}_n\mathbf{x}_t^T,$$
$$\sigma^2 = \frac{1}{\lambda^2}\mathbf{x}_t\mathbf{x}_t^T,$$
$$\text{and } \kappa(\sigma^2) = \left(1 + \sigma^2\frac{\pi}{8}\right)^{-1/2}.$$

(24)

### A.3.2 PREDICTIVE DISTRIBUTION FOR BAYESIAN NEURAL NETWORKS WITH SVGD AND DSVGD

In a manner similar to (22), the predictive distribution for BNN can be estimated as

$$p(y_t = 1|\mathbf{x}_t) \approx \sum_{n=1}^{N} \frac{1}{N(2\lambda^2\pi)^{d/2}} \int f(\mathbf{x}_t, \mathbf{w})\exp\left(\frac{-||\mathbf{w}_n - \mathbf{w}||^2}{2\lambda^2}\right)\mathrm{d}\mathbf{w},$$

(25)

where $f(\mathbf{x}_t, \mathbf{w})$ is the sigmoid output of the BNN with weights $\mathbf{w}$. Using the first order Taylor approximation of the network output around the $n$-th particle (Bishop, 2006, Sec. 5.7.1)

$$f(\mathbf{x}_t, \mathbf{w}) \approx f(\mathbf{x}_t, \mathbf{w}_n) + \nabla_{\mathbf{w}}^{\mathsf{T}}f(\mathbf{x}_t, \mathbf{w})(\mathbf{w} - \mathbf{w}_n),$$

(26)

the predictive distribution can now be rewritten as

$$p(y_t = 1|\mathbf{x}_t) \approx \sum_{n=1}^{N} \frac{1}{N(2\lambda^2\pi)^{d/2}} \int [f(\mathbf{x}_t, \mathbf{w}_n) + \nabla_{\mathbf{w}}^{\mathsf{T}}f(\mathbf{x}_t, \mathbf{w})(\mathbf{w} - \mathbf{w}_n)]\exp\left(\frac{-||\mathbf{w}_n - \mathbf{w}||^2}{2\lambda^2}\right)\mathrm{d}\mathbf{w}$$

$$= \sum_{n=1}^{N} \frac{1}{N}f(\mathbf{x}_t, \mathbf{w}_n) + \sum_{n=1}^{N} \frac{1}{N}\left(\nabla_{\mathbf{w}}^{\mathsf{T}}f(\mathbf{x}_t, \mathbf{w})\mathbf{w}_n - \nabla_{\mathbf{w}}^{\mathsf{T}}f(\mathbf{x}_t, \mathbf{w})\mathbf{w}_n\right)$$

$$= \sum_{n=1}^{N} \frac{1}{N}f(\mathbf{x}_t, \mathbf{w}_n),$$

(27)

where we have used the fact that $\int \mathcal{N}(\mathbf{w}|\mathbf{w}_n, \lambda^2\mathbf{I}_d)\mathrm{d}\mathbf{w} = 1$ and $\int \mathbf{w}\mathcal{N}(\mathbf{w}|\mathbf{w}_n, \lambda^2\mathbf{I}_d)\mathrm{d}\mathbf{w} = \mathbf{w}_n$.

### A.4 SPACE-TIME COMPLEXITY, COMMUNICATION LOAD AND CONVERGENCE

This section offers a brief discussion on the complexity, communication load and convergence of DSVGD.

**Space Complexity.** DSVGD inherits the space complexity of SVGD. In particular, DSVGD requires the computation of the kernel matrix $\mathrm{k}(\cdot, \cdot)$ between all particles at each local iteration, which

can then be deleted before the next iteration. This requires $\mathcal{O}(N^2)$ space complexity. As pointed out by Liu & Wang (2016) and noticed in our experiments, for sufficiently small problems of practical interest for mobile embedded applications, few particles are enough to obtain state-of-the art performance. Furthermore, $N$ particles of dimension $d$ need to be saved in the local buffer, requiring $\mathcal{O}(Nd)$ space. Given that $N$ is generally much lower than the number of data samples, saving the particles in the local buffer shouldn't be problematic.

**Time complexity.** When scheduled, an agent has to perform $\mathcal{O}(\max(L, L')N^2)$ operations with $\mathcal{O}(LN^2)$ operations for the first loop (lines **5-11**) and $\mathcal{O}(L'N^2)$ operations for the second loop (lines **15-21**) in Algorithm 1. Furthermore, the $L'$ distillation iterations in the second loop can be performed by the scheduled agent after it has sent its global particles to the central server. This enables the pipelining of the second loop with the operations at the server and at other agents, which can potentially reduce the wall-clock time per communication round.

**Communication load.** Using DSVGD, the communication load between a scheduled agent and the central server is of the order $\mathcal{O}(Nd)$ since $N$ particles of dimensions $d$ need to be exchanged at each communication round. In contrast, the communication load of PVI depends on the selected parametrization. For instance, one can use PVI with a fully factorized Gaussian approximate posterior, which requires only $2d$ parameters to be shared with the server, namely mean and variance of each of the $d$ parameters at the price of having lower accuracy.

**Convergence.** The two local SVGD loops produce a set of global and local particles, respectively, that are convergent to their respective targets as the number $N$ of particles increases (Liu, 2017a). Furthermore, as discussed, a fixed point of the set of local free energy minimization problems is guaranteed to be a local optimum for the global free energy problem (see Property 3 in Bui et al. (2018)). This property hence carries over to DSVGD in the limit of large number of particles. However, convergence to a fixed point is an open question for PVI, and consequently also for DSVGD.

## A.5 PARALLEL-DSVGD

In this section, we present a direct extension of DSVGD in which multiple agents can be scheduled in parallel during the same communication round. In Parallel-DSVGD (P-DSVGD), each agent in the set $\mathcal{K}^{(i)}$ of scheduled agents at round $i$ applies the same steps as in DSVGD except that it shares the local particles $\{\theta_{k,n}^{(i)}\}_{n=1}^{N}$ with the server instead of the global ones. Then, the server distills the received local particles into a set of $N$ server-side particles $\{\phi_n^{(i)}\}_{n=1}^{N}$ using SVGD to obtain the next iterate of the global posterior.

As discussed in Sec. 3, a parallel implementation requires the $i$-th iterate of the global posterior to be obtained as

$$q^{(i)}(\theta) = p_0(\theta) \prod_{k \in \mathcal{K}^{(i)}} t_k^{(i)}(\theta) \prod_{k' \notin \mathcal{K}^{(i)}} t_{k'}^{(i)}(\theta), \tag{28}$$

where $t_{k'}^{(i)}(\theta) = t_{k'}^{(i-1)}(\theta)$ for $k' \notin \mathcal{K}^{(i)}$. To replicate this same behaviour while preserving the non-parametric property of DSVGD, in P-DSVGD, each agent $k \in \mathcal{K}^{(i)}$ shares its local particles $\{\theta_{k,n}^{(i)}\}_{n=1}^{N}$ representing the approximate likelihood where $t_k^{(i)}(\theta) = N^{-1} \sum_n^N \mathrm{K}(\theta, \theta_{k,n}^{(i)})$. Then, to approximate $q^{(i)}(\theta)$ in (28), using SVGD, the server carries out $L_s$ SVGD updates as

$$\phi_n^{[l]} \leftarrow \phi_n^{[l-1]} + \frac{\epsilon}{N} \sum_{j=1}^{N} [\mathrm{k}(\phi_j^{[l-1]}, \phi_n^{[l-1]}) \nabla_{\theta_j} \log q^{(i)}(\phi_j^{[l-1]}) + \nabla_{\phi_j} \mathrm{k}(\phi_j^{[l-1]}, \phi_n^{[l-1]})], \text{for } l=1,\ldots,L_s. \tag{29}$$

For the $(i+1)$-th communication round, scheduled agents $\mathcal{K}^{(i+1)}$ download particles $\{\phi_n^{(i+1)}\}_{n=1}^{N} = \{\phi_n^{[L_s]}\}_{n=1}^{N}$ that are treated in a similar fashion as in DSVGD. The full algorithmic table for Parallel-Distributed Stein Variational Gradient Descent (P-DSVGD) is provided in Algorithm 5. Numerical results for P-DSVGD are provided in Sec. B.3 of the Appendix.

## A.6 PROOFS

In this section, we prove Theorem 1 and 2.

**Theorem 1.** *The global posterior $q_{opt}(\theta)$ in (2) is the unique fixed point of the DVI algorithm.*

*Proof.* Consider the general implementation of DVI, were a set $\mathcal{K}$ of agents are scheduled in parallel. DVI is equivalent to the following functional mapping

$$
\begin{bmatrix} \prod_{i \notin \mathcal{K}} t_i(\theta) \\ \{t_k(\theta)\}_{k \in \mathcal{K}} \end{bmatrix} \rightarrow \begin{bmatrix} \prod_{i \notin \mathcal{K}} t_i(\theta) \\ \left\{ t'_k(\theta) = \frac{1}{Z} \exp\left( -\frac{1}{\alpha} L_k(\theta) \right) \right\}_{k \in \mathcal{K}} \end{bmatrix}
$$

$$
\left( q(\theta) = p_0(\theta) \prod_{i=1}^{K} t_i(\theta) \right) \qquad \left( q'(\theta) = p_0(\theta) \prod_{i \notin \mathcal{K}} t_i(\theta) \prod_{k \in \mathcal{K}} t'_k(\theta) \right)
$$

where $Z = \int p_0(\theta) \prod_{i \notin \mathcal{K}} t_i(\theta) \prod_{k \in \mathcal{K}} t'_k(\theta) d\theta$.
Therefore, assuming that all devices $k$ are periodically scheduled, $q(\theta)$ is a fixed point of DVI if and only if the following equality holds

$$
t_k(\theta) = t'_k(\theta) \text{ for } k = 1, \ldots, K.
$$

This condition is satisfied by $q(\theta) = q_{opt}(\theta)$ and by no other distribution. This concludes the proof. □

We move now to Theorem 2 for U-DSVGD. We leave the analysis of the impact of the additional distillation step used by DSVGD for future work. The analysis builds on the following result from Korba et al. (2020), which is restated here using our notation.

Denote by $|| \cdot ||_{\mathcal{H}}$ the norm in the RKHS $\mathcal{H}$ defined by the positive definite kernel $\mathrm{k}(\theta, \theta')$. We assume that the kernel satisfies the following technical condition: there exist a constant $B > 0$ such that

$$
||\mathrm{k}(\theta, \cdot)||_{\mathcal{H}} \le B \text{ and } \sum_{j=1}^{d} \left|\left| \frac{\partial \mathrm{k}(\theta, \cdot)}{\partial \theta_j} \right|\right|_{\mathcal{H}}^2 \le B^2. \tag{30}
$$

This condition is for instance satisfied by the RBF kernel with $B = 1$ (Zhou, 2008). Furthermore, we define the kernelized Stein discrepancy (Liu et al., 2016) between two distributions $p$ and $q$ as $S(p, q)$, and the total variation distance as $||q - p||_{TV} = \frac{1}{2} \int |q(\theta) - p(\theta)| d\theta$.

**Lemma 1.** *(Guaranteed per-iteration decrease of the local free energy.)* (Korba et al., 2020) For a kernel satisfying (30), assume that, at a given communication round $i$ and local iteration $l$, with agent $k$ scheduled, we have:

- the maximum absolute eigenvalue of the Hessian $-\nabla^2 \log \tilde{p}_k^{(i)}(\theta)$ is upper bounded by a constant $M > 0$; and
- the inequality $S(q^{[l]}(\theta), \tilde{p}_k^{(i)}) < C$ holds for some $C > 0$.

For learning rate $\epsilon \le (\beta - 1)/(\beta B C^{\frac{1}{2}})$ with any $\beta > 1$, the decrease in the local KL divergence from local iteration $l$ to $l + 1$ satisfies the inequality

$$
F(q^{[l+1]}(\theta)) - F(q^{[l]}(\theta)) \le -\alpha \epsilon S(q^{[l]}, \tilde{p}_k^{(i)})(1 - \epsilon \gamma), \tag{31}
$$

where $\gamma = ((\beta^2 + M)B^2)/2$.

Lemma 1 shows that by choosing a learning rate $\epsilon \le \min(\gamma^{-1}, (\beta - 1)/(\beta B C^{\frac{1}{2}}))$, one can guarantee a per-iteration decrease in the local-free energy, i.e., in the KL divergence between the particles' distribution and the target tilted distribution $\tilde{p}_k^{(i)}(\theta)$ that depends on the kernelized Stein discrepancy $S(q^{[l]}, \tilde{p}_k^{(i)})$ at the iteration before the update.

**Lemma 2.** *(Relationship between global and local free energy.)* The global free energy $F(q(\theta))$ in (1) is related to the local free energy $F_k^{(i)}(q(\theta))$ in (4) of the $k$-th scheduled agent as

$$F(q(\theta)) = F_k^{(i)}(q(\theta)) + \alpha \sum_{m \neq k} \mathbb{E}_{q(\theta)} \log \left( \frac{t_m^{(i-1)}(\theta)}{\exp(-\frac{1}{\alpha} L_m(\theta))} \right). \tag{32}$$

*Proof.* The global free energy (1) can be written as

$$
\begin{aligned}
F(q(\theta)) &= \alpha \mathbb{E}_{q(\theta)} \log \left( \frac{q(\theta)}{p_0(\theta) \exp(-\frac{1}{\alpha} \sum_{m=1}^{K} L_m(\theta))} \right) \\
&= \alpha \mathbb{E}_{q(\theta)} \log \left( \frac{q(\theta)}{\tilde{p}_k^{(i)}(\theta)} \cdot \frac{\frac{q^{(i-1)}(\theta)}{t_k^{(i-1)}(\theta)}}{p_0(\theta) \exp(-\frac{1}{\alpha} \sum_{m \neq k} L_m(\theta))} \right) \\
&= \alpha \mathbb{E}_{q(\theta)} \log \left( \frac{q(\theta)}{\tilde{p}_k^{(i)}(\theta)} \right) + \alpha \mathbb{E}_{q(\theta)} \log \left( \frac{p_0(\theta) \prod_{m \neq k} t_m^{(i-1)}(\theta)}{p_0(\theta) \exp(-\frac{1}{\alpha} \sum_{m \neq k} L_m(\theta))} \right) \\
&= F_k^{(i)}(q(\theta)) + \alpha \sum_{m \neq k} \mathbb{E}_{q(\theta)} \log \left( \frac{t_m^{(i-1)}(\theta)}{\exp(-\frac{1}{\alpha} L_m(\theta))} \right),
\end{aligned}
\tag{33}
$$

where in the second equality we have used (11); and in the third equality we have used the equality $q^{(i-1)}(\theta) = p_0(\theta) \prod_{m=1}^{K} t_m^{(i-1)}(\theta)$, which is guaranteed by the U-DSVGD update (10) and (11) (see Bui et al. (2018, Property 2)). $\qquad \square$

**Theorem 2** (Guaranteed per-iteration decrease of the global free energy.)**.** *The decrease in the global free energy from local iteration $l$ to $l+1$ during communication round $i$ for which agent $k$ is scheduled can be lower bounded as*

$$F(q^{[l]}(\theta)) - F(q^{[l+1]}(\theta)) \geq \alpha \epsilon S(q^{[l]}, p_k^{(i)})(1 - \epsilon \gamma) - 2\alpha(K-1) l_{\max}^{(i)} \sqrt{2\mathbb{D}(q^{[l+1]} || q^{[l]})}, \tag{16}$$

*where $l_{\max}^{(i)} = \sup_{\theta} \max_{m \neq k} |\log(t_m^{(i-1)}(\theta)) \cdot \exp(\frac{1}{\alpha} L_m(\theta))|$, $S(q, p)$ denotes the Kernalized Stein Discrepancy between distributions $q$ and $p$ (Liu et al., 2016), and $\gamma$ is a constant depending on the RKHS kernel and the target distribution.*

We know from Lemma 1 that a learning rate $\epsilon \leq \min(\gamma^{-1}, (\beta - 1)/(\beta BC^{\frac{1}{2}})$ is sufficient to ensure a per-iteration decrease in the *local* free energy. Given that the KL divergence in the second term in (16) generally increases with $\epsilon$, 2 demonstrates that, in order to guarantee a reduction of the *global* free energy, a smaller learning rate may be required. We also note that the KL divergence term $\mathbb{D}(q^{[l+1]} || q^{[l]})$ may be explicitly related to the learning rate by following Pinder et al. (2020, Sec. 8), but we do not further pursue this aspect here. We finally remark that, in the presence of $K = 1$ agent, the upper bound (31) in (Korba et al., 2020) is recovered. This is because, in the presence of one agent, the global free energy reduces to the local free energy (see (32)) and accordingly U-DSVGD reduces to SVGD.

*Proof.* We wish to obtain an upper bound on the decrease of the global free energy $F(q^{[l+1]}(\theta)) - F(q^{[l]}(\theta))$ across each local SVGD iteration during communication round $i$. Using (32), the decrease in the global free energy can be written as

$$
\begin{aligned}
F(q^{[l+1]}(\theta)) - F(q^{[l]}(\theta)) = &\underbrace{F_k^{(i)}(q^{[l+1]}(\theta)) - F_k^{(i)}(q^{[l]}(\theta))}_{(a)} \\
&+ \alpha \sum_{m \neq k} \bigg[ \underbrace{\mathbb{E}_{q^{[l+1]}(\theta)} \log \left( \frac{t_m^{(i-1)}(\theta)}{\exp(-\frac{1}{\alpha} L_m(\theta))} \right) - \mathbb{E}_{q^{[l]}(\theta)} \log \left( \frac{t_m^{(i-1)}(\theta)}{\exp(-\frac{1}{\alpha} L_m(\theta))} \right)}_{(b)} \bigg].
\end{aligned}
\tag{34}
$$

We now derive upper bounds for $(a)$ and $(b)$. Using Lemma 1 and the definition of the local free energy in (4), we have the following upper bound on $(a)$

$$(a) = F_k^{(i)}(q^{[l+1]}(\theta)) - F_k^{(i)}(q^{[l]}(\theta)) \leq -\alpha \epsilon S(q^{[l]}(\theta), \tilde{p}_k^{(i)}(\theta))(1 - \epsilon \gamma), \tag{35}$$

while $(b)$ can be rewritten and upper bounded by using the properties of the total variation distance as

$$(b) = \int (q^{[l+1]}(\theta) - q^{[l]}(\theta)) \log \left( \frac{t_m^{(i-1)}(\theta)}{\exp(-\frac{1}{\alpha} L_m(\theta))} \right) d\theta \leq 2l_{\max}^{(i)} ||q^{[l+1]} - q^{[l]}||_{TV}. \tag{36}$$

Using Pinsker's inequality (Pinsker, 1964), the term $(b)$ can be further upper bounded as

$$(b) \leq 2l_{\max}^{(i)} \sqrt{2\mathbb{D}(q^{[l+1]}||q^{[l]})}. \tag{37}$$

Accordingly, the global energy dissipation in (34) can be upper bounded as in (16). $\qquad\square$

# B  ADDITIONAL EXPERIMENTS

An overview of the benchmarks considered in the experiments is provided in Table 1.

Table 1: Overview of benchmarks used in the experiments.

| Algorithm | Non-parametric | Decentralized | Inference |
|---|---|---|---|
| Stein Variational Gradient Descent (SVGD) (Liu & Wang, 2016) | **Yes** | No | VI |
| Stochastic Gradient Langevin Dynamics (SGLD) (Welling & Teh, 2011) | **Yes** | No | MC |
| Distributed Stochastic Gradient Langevin Dynamics (DSGLD) (Ahn et al., 2014) | **Yes** | **Yes** | MC |
| Particle Mirror Descent (PMD) (Dai et al., 2016) | **Yes** | No | VI |
| Partitioned Variational Inference (PVI) (Bui et al., 2018) | No | **Yes** | VI |
| Global Variational Inference (GVI) (Sato, 2001) | No | No | VI |
| Non-Parametric Variational Infernce (NPV) (Gershman et al., 2012) | No | No | VI |
| Federated Averaging (FedAvg) (McMahan et al., 2017) | No | **Yes** | Freq. |
| Federated Stochastic Gradient Descent (FedSGD) (McMahan et al., 2017) | No | **Yes** | Freq. |
| **Distributed Stein Variational Gradient Descent (DSVGD)** (ours) | **Yes** | **Yes** | VI |

T

## B.1  1-D MIXTURE OF GAUSSIANS TOY EXAMPLE

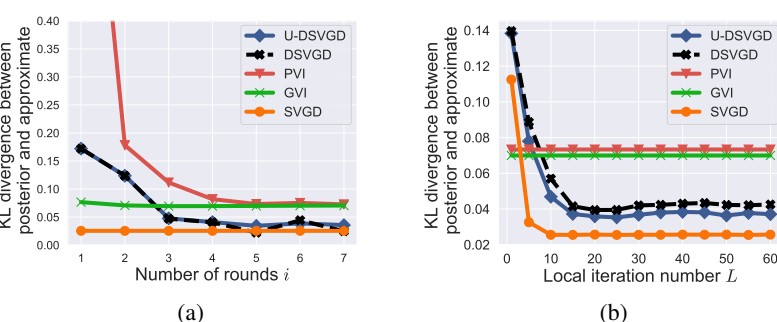

(a)  (b)

Figure 9: KL divergence between exact and approximate global posteriors (a) as function of the number of communication rounds $i$ for $L = L' = 200$; and (b) as function of the local iterations number $L$ for $I = 5$.

This section is complementary to the 1-D mixture of Gaussians experiment in Sec. 7 of the main text. We compare DSVGD with PVI and the counterpart centralized schemes. In Fig. 9(a), we plot the KL divergence between the global posterior $q_{opt}(\theta)$ and its current approximation $q(\theta)$ as a function of the number of communication rounds $i$, which corresponds to the number of communication rounds for decentralized schemes. We use $N = 200$ particles for U-DSVGD and DSVGD with $L = L' = 200$ local iterations. The number of SVGD iterations is fixed to $800$. A Gaussian prior $p_0(\theta) = \mathcal{N}(\theta|0, 1)$ is assumed in lieu of the uniform prior considered in Fig. 2 to facilitate the implementation of PVI and conventional centralized GVI which was done following Bui et al. (2018, Property 4). More specifically, we use Gaussian approximate likelihoods, i.e., $t_k(\theta|\eta) = \mathcal{N}(\theta|\frac{-\eta_1}{2\eta_2}, \frac{-1}{2\eta_2})$ with natural parameters $\eta_1$ and $\eta_2 < 0$. We observe that DSVGD has similar convergence speed as PVI, while having a superior performance thanks to the reduced bias of non-parametric models. Furthermore, DSVGD exhibits the same performance as U-DSVGD with the advantage of having memory requirements that do not scale with the number of iterations. Finally, both U-DSVGD and DSVGD converge to the performance of (centralized) SVGD as the number of rounds increases.

In Fig. 9(b), we plot the same KL divergence as function of the number of local iterations $L$. We use $I = 5$ rounds for the decentralized schemes. It is observed that non-parametric schemes-namely SVGD and (U-)DSVGD-require a sufficiently large number of local iterations in order to outperform the parametric strategies PVI and GVI.

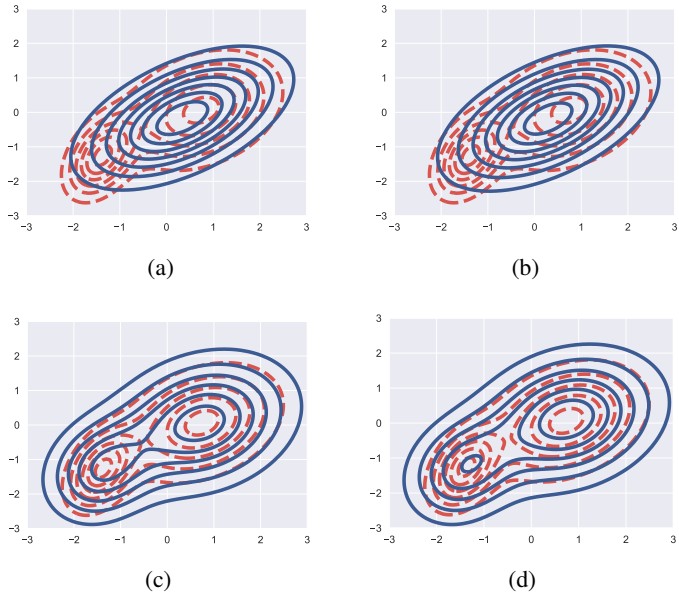

Figure 10: Performance comparison of (a) GVI, (b) PVI, (c) SVGD and (d) DSVGD for a multivariate Gaussian mixture model. Solid contour lines correspond to the approximate posterior while dashed contour lines to the exact posterior ($N = 200$, $I = 5$, $L = 200$ and $b = 0.1$).

### B.2    2-D Mixture of Gaussians Toy Example

We now consider the following 2-D mixture of Gaussians model: $p_1(\theta) = \mathcal{N}(\boldsymbol{\mu}_0, \boldsymbol{\Sigma}_0)(\mathcal{N}(\boldsymbol{\mu}_1, \boldsymbol{\Sigma}_1) + \mathcal{N}(\boldsymbol{\mu}_2, \boldsymbol{\Sigma}_2))$ and $p_2(\theta) = \mathcal{N}(\boldsymbol{\mu}_0, \boldsymbol{\Sigma}_0)\mathcal{N}(\boldsymbol{\mu}_3, \boldsymbol{\Sigma}_3)$ where

$$\boldsymbol{\mu}_0 = [0, 0] \; ; \; \boldsymbol{\Sigma}_0 = \begin{bmatrix} 4 & 2 \\ 2 & 4 \end{bmatrix}$$

$$\boldsymbol{\mu}_1 = [-1.71, -1.801] \; ; \; \boldsymbol{\Sigma}_1 = \begin{bmatrix} 0.226 & 0.1652 \\ 0.1652 & 0.6779 \end{bmatrix}$$

$$\boldsymbol{\mu}_2 = [1, 0] \; ; \; \boldsymbol{\Sigma}_2 = \begin{bmatrix} 2 & 0.5 \\ 0.5 & 2 \end{bmatrix}$$

$$\boldsymbol{\mu}_3 = [1, 0] \; ; \; \boldsymbol{\Sigma}_3 = \begin{bmatrix} 3 & 0.5 \\ 0.5 & 3 \end{bmatrix}.$$

We plot in Fig. 10 the approximate posterior $q(\theta)$ (blue solid contour lines) and the exact posterior $q_{opt}(\theta)$ (red dashed contour lines) for PVI, GVI, SVGD and DSVGD. We see that, as in the 1-D case and in contrast to parametric methods PVI and GVI, non-parametric methods SVGD and DSVGD are able to capture the different modes of the posterior, obtaining lower values for the KL divergence between the approximate and exact posterior.

### B.3    Bayesian Logistic Regression

This section provides additional results for the Bayesian logistic regression experiment in Sec. 7 of the main text. In Fig. 11, we compare the performance of DSVGD (bottom row), and U-DSVGD (top row) both with SVGD and NPV (Gershman et al., 2012) using the model described in Sec. 7. We use 9 binary classification datasets summarized in Appendix C as used in Liu & Wang (2016) and Gershman et al. (2012). We assumed $N = 100$ particles. To ensure fairness, we used $L = 800$ iterations for SVGD, while U-DSVGD and DSVGD are executed with two agents with half of the dataset split randomly at each agent. We set $I = 4$ rounds and $L = L' = 200$ local iterations. In Fig. 11, we plot the accuracy and the log-likelihood of the four algorithms. We observe that both U-DSVGD and DSVGD perform similarly to SVGD and NPV over most datasets, while allowing a

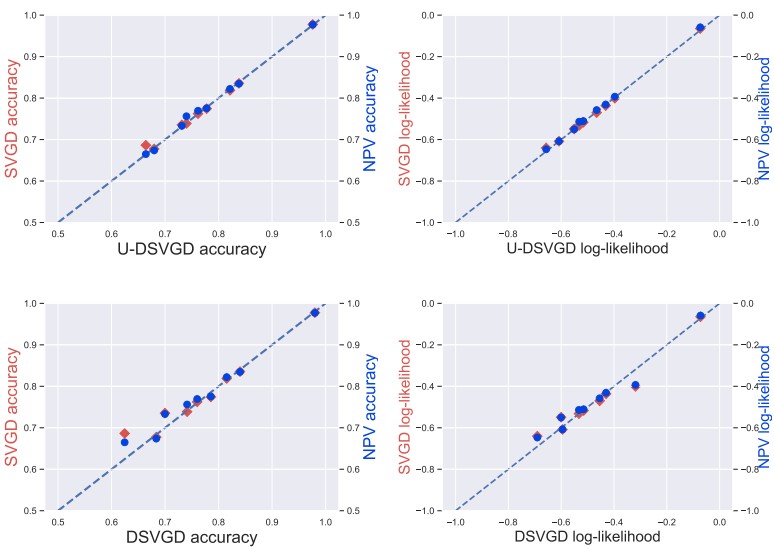

Figure 11: Binary classification with Bayesian logistic regression results using the setting in Gershman et al. (2012): accuracy and log-likelihood for U-DSVGD (upper row) and DSVGD (bottom row), along with NPV and SVGD, for various datasets.

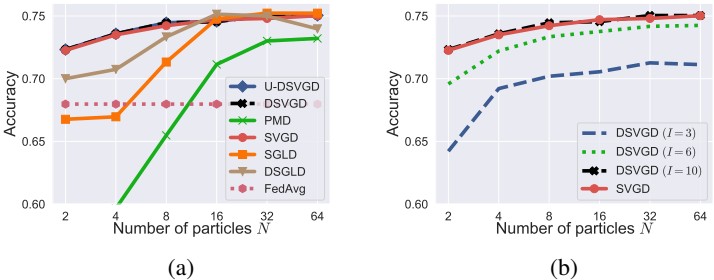

Figure 12: Accuracy as a function of the number of particles $N$ for Bayesian logistic regression on the Covertype dataset: (a) comparison with various benchmarks summarized in Table 1, and (b) performance for different number of rounds $I$. $L = 2000$ iterations were used for centralized schemes while $I \times L = 10 \times 200$ total local iterations were used for decentralized schemes.

distributed implementation. We note that NPV requires computation of the Hessian matrix which is relatively impractical to compute.

We plot in Fig. 12(a) the accuracy as function of the number of particles $N$. DSGLD is executed with two agents, where $N/2$ chains per agent are ran for a trajectory of length $4$ and $500$ rounds, which we have found to work best. We found that SVGD, DSVGD and U-DSVGD exhibit the same performance, which is superior to Particle Mirror Descent (PMD) and similar to SGLD and DSGLD when the number of particles increases. Fig. 12(b) plots the accuracy for DSVGD for the same setting for different number of communication rounds. We can see that, by increasing the number of particles, i.e., the communication load, one can obtain similar accuracy as for a lower number of particles but with a higher number of communication rounds. For example, $N = 8$ with $I = 6$ communication rounds achieves similar performance as $N = 4$ with $I = 10$ communication rounds.

Fig. 13 is a complementary figure for Fig. 4 in the main text. It shows that similar conclusions based on accuracy can be made when using the log-likelihood.

Fig. 14 shows the accuracy of DSVGD for different datasets as function of the total number $L$ of local iterations. We fix $N = 6$, $I = 10$, $L = L' = 200$ for U-DSVGD, DSGLD and DSVGD while $L = 2000$ for SVGD and SGLD. We observe that U-DSVGD and DSVGD have similar performance to SVGD and that they consistently outperform other schemes for sufficiently high $L$.

Fig. 16 is complementary to Fig. 4 in the main text. We note that the slightly noisy behaviour of DSVGD with $K = 20$ agents is attributed to the small local dataset sizes resulting from splitting the original small datasets.

Finally, Fig. 15 compares the accuracy of P-DSVGD with FedAvg and DSGLD with $K = 100$ agents and a proportion of $0.2$ randomly scheduled agents per communication round. We see that P-DSVGD exhibits similar behaviour and gain over other schemes similarly to DSVGD.

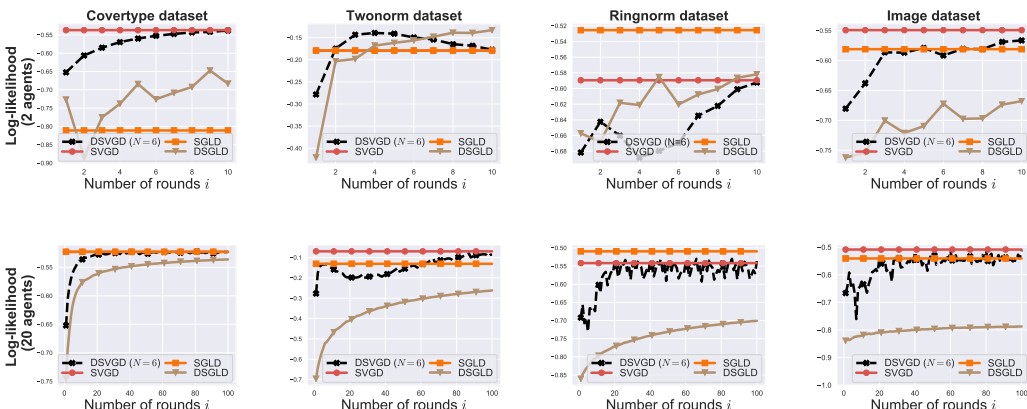

Figure 13: Bayesian logistic regression log-likelihood with $K = 2$ and $K = 20$ agents using the setting in Gershman et al. (2012) comparing DSVGD to distributed (DSGLD) and centralized (SVGD and SGLD) schemes as function of the number of communication rounds $i$. We use $N = 6$ particles and fix $L = L' = 200$. FedAvg has been removed as it has a log-likelihood lower than $-1$ in all cases and to allow us to focus on relevant values for DSVGD.

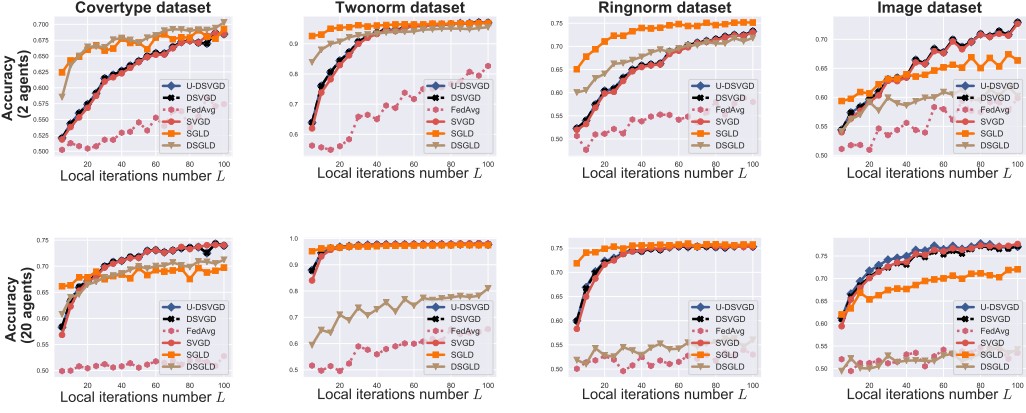

Figure 14: Bayesian logistic regression accuracy for $K = 2$ (top row) and $K = 20$ (bottom row) agents using the setting in Gershman et al. (2012) comparing U-DSVGD and DSVGD to distributed (DSGLD) and centralized (SVGD and SGLD) schemes as function of the local iterations number $L$. We fix $N = 6$ particles, $I = 5$ (top row) and $I = 20$ (bottom row).

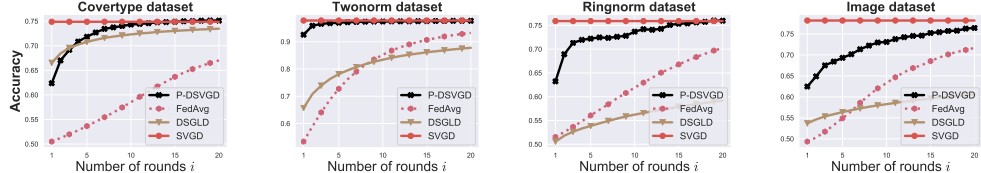

Figure 15: Accuracy for Bayesian logistic regression with P-DSVGD, FedAvg, DSGLD and SVGD using different datasets with $K = 100$ agents and a proportion of $C = 0.2$ randomly scheduled agents. SVGD was executed for $L = C \times 100 \times 4000$ iterations while we fix $L = L' = L_s = 200$ total local iterations for the remaining schemes. We use $N = 6$ particles.

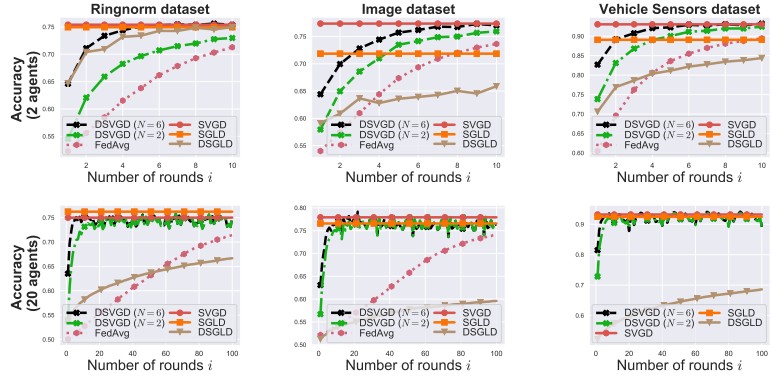

Figure 16: Accuracy for Bayesian logistic regression with $K = 2$ (top row) and $K = 20$ (bottom row) agents under the setting in Gershman et al. (2012) as function of the number of communication rounds $i$, or number of communication rounds ($N = 6$ particles, $L = L' = 200$).

### B.4 BAYESIAN NEURAL NETWORKS FOR REGRESSION AND CLASSIFICATION

This part contains additional results on regression and multilabel classification experiments using Bayesian Neural Networks. Figures 17 and 18 are complementary to Figures 6 and 7 in the main text and validate our conclusions using additional datasets for regression and the log-likelihood metric for multi-label classification.

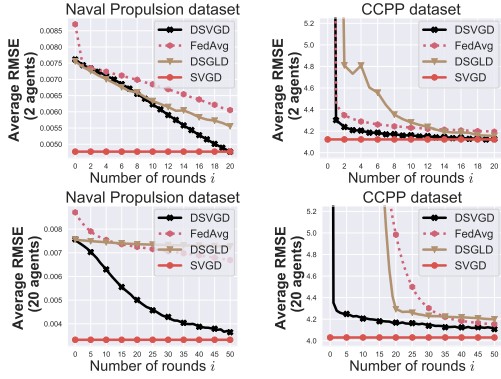

Figure 17: Average Root Mean Square Error (RMSE) as a function of the number of communication rounds $i$, or number of communication rounds, for regression using Bayesian neural networks with a single hidden layer of ReLUs under the setting of Hernández-Lobato & Adams (2015), with $K = 2$ (top row) and $K = 20$ (bottom row) agents. ($N = 20$, $L = L' = 200$ and $50$ hidden neurons).

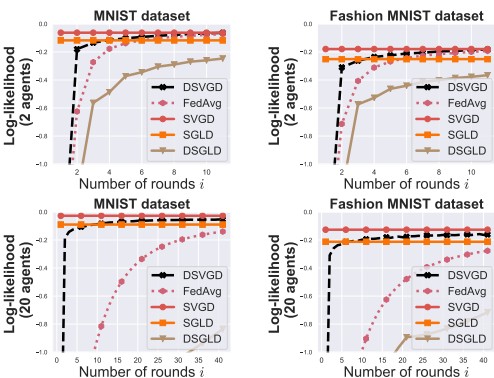

Figure 18: Log-likelihood for multi-label classification using Bayesian neural networks with a single hidden layer of 100 neurons as function of the number of communication rounds $i$, or number of communication rounds, using MNIST and Fashion MNIST with $K = 2$ (top row) and $K = 20$ (bottom row) agents ($N = 20$, $L = L' = 200$.

## B.5 RELIABILITY PLOTS AND MAXIMUM CALIBRATION ERROR

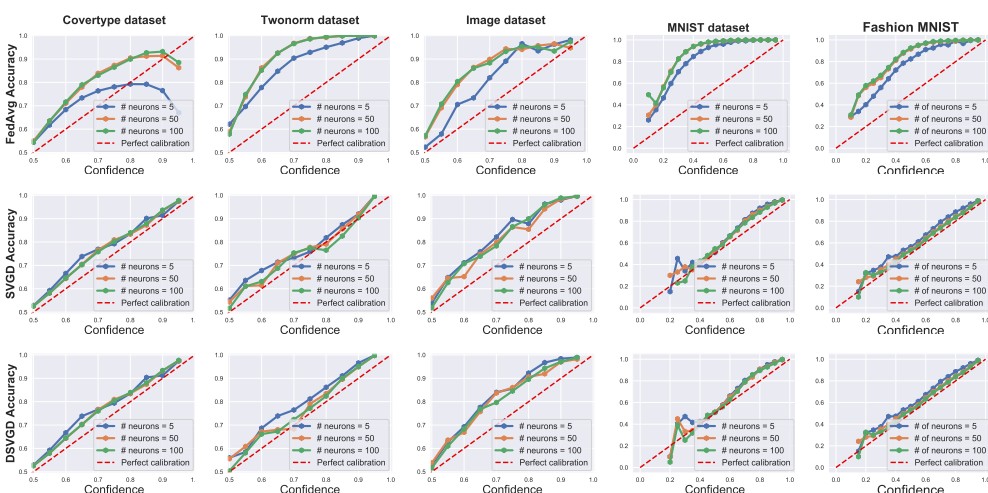

Figure 19: Reliability plots for classification using Bayesian neural networks for a variable number of hidden neurons with FedAvg (top row), SVGD (middle row) and DSVGD (bottom row). We use $N = 20$ particles ($I = 10$, $L = L' = 200$ and $K = 20$ agents).

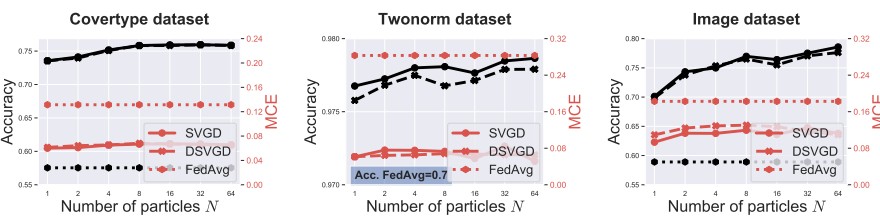

Figure 20: Accuracy and Maximum Calibration Error (MCE) as function of the number of particles $N$ for Bayesian neural networks. We fix $I = 10$, $L = L' = 200$ and $K = 20$ agents in both figures.

This section provides additional results on the calibration experiment conducted in Sec. 7 of the main text using additional datasets. In Fig. 19, we show the reliability plots for SVGD, DSVGD and FedAvg with $K = 20$ agents across various datasets and for different number of neurons in the hidden layer. We first note that DSVGD retains the same calibration level as SVGD across all datasets. Furthermore, while increasing the number of hidden neurons negatively affects FedAvg due to overfitting, it does not affect the trustworthiness of the predictions for the Bayesian counterparts. This is a general property for Bayesian methods that contrast with frequentist approaches, for which increasing the number of parameters improves accuracy at the price of miscalibration (Guo et al., 2017).

Fig. 20 plots the accuracy and MCE as function of the number of particles $N$. While increasing $N$ improves the accuracy (as also shown in Fig. 12) for SVGD and DSVGD, the MCE is unaffected and is lower than the MCE value for FedAvg.

## C    Implementation Details

### C.1    Datasets, Benchmarks and Hyperparamters Details

**Datasets.** We summarize in Table 2 the main parameters used across different datasets that are invariant across all experiments. The covertype dataset[1] and the remaining binary classfication datasets that are selected from the Gunnar Raetsch's Benchmark datasets[2] as compiled by Mika et al. (1999) are used directly without normalization as in Liu & Wang (2016) except for the vehicle sensors dataset[3] which is normalized by removing the mean of each feature and dividing by their standard deviations. Regression datasets[4] are normalized by removing the mean of each feature and dividing by their standard deviations, and multi-label classification datasets[56] are normalized by multiplying each pixel value by $0.99/255$ and adding $0.01$ such that every pixel value after normalization belongs to the interval $[0.01, 1]$. All performance metrics used are averaged over the number of trials. In each trial, unless specified otherwise, we permute the datasets and randomly split them across different agents.

**Hyperparameters.** The hyperparameters used are summarized in Table 3. These apply for all schemes except for DSGLD and SGLD, where the learning rates are annealed and are respectively equal to $a_0 \cdot (0.5 + i \cdot L + l)^{-0.55}$ and $a_0 \cdot (0.5 + l)^{-0.55}$ to ensure that they go from the order of $0.01$ to $0.0001$ as advised by Welling & Teh (2011). $a_0$ is fixed according to the values in Table 4.

**DSGLD implementation.** DSGLD is implemented by splitting the $N$ particles among the $K$ agents. More specifically, when scheduled, each agent runs $\lceil N/K \rceil$ Markov chains. We assumed that the response delay in addition to the trajectory length of the chains (Ahn et al., 2014) to be equal among all workers and unchanged throughout the learning process.

**FedAvg implementation.** FedAvg is implemented as in McMahan et al. (2017) with the only difference that the server schedules a single agent at a time. Each scheduled agent performs $L$ SGD iterations to minimize its local loss.

**PVI and GVI implementation.** PVI and GVI are implemented using a Gaussian parametrization for both the posterior and the prior. The natural parameters are updated via the closed form update in Bui et al. (2018, Property 4).

**Scheduling.** Unless specified otherwise, we use a round robin scheduler to schedule agents. However, any scheduler can be used as long as it schedules one agent per communication round.

Table 2: Overview of datasets and parameters used in the experiments. Datasets in bold are used in the experiments section of the main text.

| Dataset Name | Size | Task | batchsize | # trials | Train/test split |
|---|---|---|---|---|---|
| **Covertype** | $581,012 \times 55$ | Binary classification | 100 | 50 | 80%/20% |
| **Twonorm** | $7,400 \times 20$ | Binary classification | 10 | 50 | 80%/20% |
| Ringnorm | $7,400 \times 20$ | Binary classification | 10 | 50 | 80%/20% |
| Image | $2,086 \times 18$ | Binary classification | 10 | 50 | 80%/20% |
| Breast Cancer | $263 \times 9$ | Binary classification | 10 | 50 | 80%/20% |
| Diabetis | $768 \times 8$ | Binary classification | 10 | 50 | 80%/20% |
| German | $1,000 \times 20$ | Binary classification | 10 | 50 | 80%/20% |
| Heart | $270 \times 13$ | Binary classification | 10 | 50 | 80%/20% |
| Waveform | $5,086 \times 21$ | Binary classification | 10 | 50 | 80%/20% |
| Vehicle Sensors | $2010 \times 23$ | Binary classification | 10 | 50 | 80%/20% |
| **Kin8nm** | $8,192 \times 8$ | Regression | 100 | 50 | 90%/10% |
| Naval Propulsion | $11,934 \times 16$ | Regression | 100 | 50 | 90%/10% |
| Combined cycle power plant (CCPP) | $9,568 \times 4$ | Regression | 100 | 50 | 90%/10% |
| **Year Prediction** | $515,345 \times 90$ | Regression | 1000 | 20 | 90%/10% |
| **MNIST** | $60,000 \times 785$ | Multi-label classification | 100 | 20 | 86%/14% |
| **Fashion MNIST** | $60,000 \times 785$ | Multi-label classification | 100 | 20 | 86%/14% |

---

[1] https://www.csie.ntu.edu.tw/~cjlin/libsvmtools/datasets/binary.html
[2] http://theoval.cmp.uea.ac.uk/matlab/default.html
[3] http://www.ecs.umass.edu/~mduarte/Software.html
[4] https://archive.ics.uci.edu/ml/datasets.php
[5] http://yann.lecun.com/exdb/mnist/
[6] https://github.com/zalandoresearch/fashion-mnist

Table 3: Summary of hyperparameters used across various experiments.

| Hyperparameter | Regression | Binary Classification | Multi-label Classification |
|---|---|---|---|
| Ada Learning rate[7] | 0.001 | 0.05 | 0.001 |
| Ada smoothing term (or fuge factor) | $10^{-6}$ | $10^{-9}$ | $10^{-6}$ |
| Momentum | 0.9 | 0.9 | 0.9 |
| KDE bandwidth | 0.55 | 0.55 | 0.55 |

Table 4: Learning rate for DSGLD and SGLD used across various datasets.

| Hyperparameter | Year | MNIST | F-MNIST | Other |
|---|---|---|---|---|
| DSGLD $a_0$ | 0.0005 | 0.0005 | 0.0005 | 0.01 |
| SGLD $a_0$ | - | 0.001 | 0.001 | 0.01 |

## C.2 SOFTWARE DETAILS

We implement all experiments in PyTorch (Paszke et al., 2019) Version 10.3.1. Our experiments and code are based on the original SVGD experiments and code available at: `https://github.com/DartML/Stein-Variational-Gradient-Descent`. More specifically, DSVGD can be easily obtained by running SVGD twice at each scheduled agent and suitably adjusting its target distribution. Our code is attached with the supplementary materials.

---

[7]All learning rates for non-parametric particle-based benchmark schemes used are scaled by a factor of $1/N$ to match our learning rate and ensure fair comparison.

