# OpenReview forum: "Federated Generalized Bayesian Learning via  Distributed Stein Variational Gradient Descent"
_ICLR.cc/2021/Conference — Reject_

### Official Review · AnonReviewer4 · 2020-10-28
**Promising but incomplete federated learning algorithm**

**Rating:** 6
**Confidence:** 3

**Review:**

Promising but incomplete federated learning algorithm

This paper proposes a federated version of the Stein Variational Gradient Descent (SVGD) method. The general approach to perform federated learning is based on a previously published method called Partitioned Variational Inference (PVI). This work takes the PVI approach and adapts it to the SVGD framework.

The paper is in general well-written and easy to follow. Main ideas are clearly highlighted, and the technical parts are well structured and provide enough details.  The study problem is of great relevance because most of the data today is generated and stored in a distributed way. The presented approach is sound and builds on top of well-established methods.

Strong points:

- The paper addresses a very relevant problem by combining two well-founded approaches.

- The use of particle-based variational inference methods in the context of federated learning is worth exploring.

- The presented approach is rigorous and well evaluated.

- The presented approach can train models with similar prediction performance than standard centralized approaches, and it's also able to produce well-calibrated predictions.

Weak points:

- The presented approach has limited practical use because of the current restrictions it imposes (i.e. update one agent at a time).

- The convergence of the approach seems quite dubious once the current constrains of "one agent updated at a time" is lifted.

- The implementation of this method for federated learning of large deep neural networks cast doubts in the feasibility of the approach due to the high overhead of sending/receiving multiple set of weights.


I can not recommend the acceptation of this work for the following reasons:

- The originality of method is low because it directly builds on top of two well-established approaches PVI and SVGD. However, the combination of these two approaches is not straightforward and shows how particle-based approximation methods can be also used in this challenging setting.

- In my opinion, there is a relevant limitation to this approach which, although acknowledged by authors, is not properly discussed: Federated updates can only be done by one of the agents at a time, which implies that this particular algorithm is of limited practical use. One of the key points in federated learning is the possibility to exploit distributed computing infrastructure such as our mobile phones. So, updating one agent at a time practically makes unfeasible this possibility.

- Another limitation is the convergence of the algorithm or, at least, an iterative improvement of the "global free energy". This current version of the paper guarantees that the global free energy is decreased at every round, mainly because the algorithm only updates one agent at a time and it essentially works as the standard SVGD. The proof of convergence is directly borrowed from (Korba et al., 2020). But I have strong doubts that this approach can provide this guarantee once the "updating one agent at a time" constraint is lifted. Because the PVI framework, which is the basis of this approach, can not guarantee a decrease of the global energy or convergence at every updating round. This is not properly discussed in the paper.

- Another relevant limitation, which is inherent to the  SVGD method, is the number of particles to be used. Each particle, in the context of the deep neural networks, corresponds to the whole set of weights of the network. So, the transmission of several set of weights can lead to very significant communication costs/delays. This is not properly/explicitly discussed in the paper.



Minor comments:
Eq (17) is out of margin

Post-Rebutal: I really thank the authors for their efforts  following my comments. I think they have really addressed my concerns. I therefore raise the score of the paper and recommend it for acceptance.

---

> ### Author Response · Authors · 2020-11-16
> **Reply to reviewer (1/2)**
>
> We thank the reviewer for the time taken to write such a detailed assessment and for the useful suggestions offered to improve our work. By addressing your concerns, we believe that we have significantly improved the quality of the manuscript. Please see below for our reply.
>
> $\bullet$ The presented approach has limited practical use because of the current restrictions it imposes (i.e. update one agent at a time)
>
> This is a very important point that we had left as a future work in the conclusion of the original submission. Another reviewer also mentioned this aspect too, so we re-iterate our reply here. It is possible to schedule multiple agents in parallel, and we have investigated this direction in the revised paper. To elaborate, consider first the unconstrained optimization of the variational posterior, which we cover in Sec. 3 of the revised paper. In this case, scheduled agents $k \in \mathcal{K}^{(i)}$ in round $i$ share their local approximate likelihood’s parameter $\eta_{k}^{(i)}$, and then the server construct the global approximate posterior as
>
> $ q^{(i)}(\theta) = p_0(\theta) \prod_{k \in \mathcal{K}^{(i)}} t_{k}^{(i)}(\theta) \prod_{k^\prime \not\in \mathcal{K}^{(i)}} t_{k^\prime}^{(i)}(\theta)$, (1)
>
> where $t_{k^\prime}^{(i)} (\theta) = t_{k^\prime}^{(i-1)} (\theta) $ and $t_{k}^{(i)} (\theta) $ are updated as in equation (7) of the revised paper.
>
> At a theoretical level, the performance of the resulting scheme benefits from the same fixed-point property of the scheme that schedules a single device in each communication round. In particular, as we detail in Theorem 1 of the revised paper, the unique fixed point of this procedure is the global optimal posterior.
>
> Based on this theoretical insight, in Sec A.5 of the Appendix of the revised paper, we introduce a parallel extension of DSVGD, termed Parallel-DSVGD (P-DSVGD). In P-DSVGD, during the $i$-th communication round, scheduled agents report the local particles $\theta_{k,n}^{(i)} $, encoding the approximate likelihoods $t_{k}^{(i)}(\theta)$, for $k \in \mathcal{K}^{(i)}$, to the server. Then, to approximate $q^{(i)}(\theta)$ as in (1), we propose the use of server side particles $\phi_{n}^{(i)}$. The latter are updated via SVGD to approximate (1). Then, the new set of scheduled agent $\mathcal{K}^{(i+1)}$ download the updated $\phi_{n}^{(i+1)}$ and carries out the same steps as in DSVGD.
>
> Sec. A.5 of the Appendix reports also corresponding experimental results on several datasets.

---

> > ### Author Response · Authors · 2020-11-16
> > **Reply to reviewer (2/2)**
> >
> > $\bullet$ The implementation of this method for federated learning of large deep neural networks cast doubts in the feasibility of the approach due to the high overhead of sending/receiving multiple set of weights.
> >
> > Thanks for noting this important point. We re-iterate here based on our reply to reviewer 2 who also mentioned this issue.
> > As we point out in the revised Sec. 1 of the paper, federated learning applies at different scales, from industrial data silos to mobile devices, and each setting yields different types of challenges [9]. We are specifically interested in the small-scale federated learning regime consisting of mobile devices, each having a limited data set and running a small to average sized model due to memory constraints. Consider for instance the application of classifying activity – such as jogging, walking, and writing – based on data from an accelerometer [10], or the implementation of a health care monitor based on data from smart-watch ECG data [11]. A model with a limited number of parameters – say a few hundreds of weights – would be sufficient for the problem at hand, and each device would have a small number of data points.
> >
> > In this context, non-iid data, communication cost bottleneck, and limited computational resources, are all relevant, but one also need to consider the following additional challenges:
> >
> > •	Trustworthiness: Frequentist approaches are badly calibrated [7], as shown in Fig. 8 of our paper, which makes their use in critical applications domain such as healthcare monitoring, ill advised. This problem has been largely overlooked in the federated learning community;
> >
> > •	Number of communication rounds: When models are small, the payload per communication round may not be the main contributor to the overall latency of the training process. In contrast, accommodating many communication rounds requiring arbitrating channel access among multiple devices and may yield slow clock time convergence.
> >
> > We believe that the proposed approach successfully addresses both these challenges for the first time. In the paper, we show that it provides well-calibrated decisions and it can vastly improve the convergence speed in terms of communication rounds. It is true that each communication round requires a larger communication load since multiple particles need to be exchanged. While this is certainly a limitation for larger-scales implementations of federated learning, we argue that, for smaller-scale deployments, the trade-off between per-iteration communication load and number of communication rounds is well worth exploring and practically relevant. To see this, as an example, consider the accuracy using the covertype dataset in Fig. 3 on the right-hand side. The reach an accuracy of 70% using DSVGD with $N=6$ particles, only $5$ communication rounds are required. This means that the total communication load in terms of particles exchanged for DSVGD is $5\times6 = 30$, while, for FedAvg, around $100$ rounds are needed to reach the same accuracy, thus making the total communication load equal to $100$, which is much more than the communication load of DSVGD. We have further clarified this in Sec. 5.
> >
> > $\bullet$ Guarantee on the improvement of the global free energy once the parallel updates are used
> >
> > We have experimentally verified the performance of P-DSVGD on several datasets. Please note that the newly derived fixed-point theorem in Theorem 1 is a general result that encompasses both parallel and sequential scheduling. As other Expectation Propagation-like schemes, including PVI and VIRTUAL [6], convergence guarantees are still open problems. However, these have demonstrated excellent performance experimentally.
> >
> > Thank you.
> >
> > **References:**
> >
> > [1] Kairouz P, McMahan HB, Avent B, Bellet A, Bennis M, Bhagoji AN, Bonawitz K, Charles Z, Cormode G, Cummings R, d'Oliveira RG. Advances and open problems in federated learning. arXiv preprint arXiv:1912.04977. 2019 Dec 10.
> >
> > [2] Mary M Weiss, Kenichi Yoneda, and Thaier Hayajneh, “Smartphone and smartwatch-based biometrics using activities of daily living,” IEEE Access, vol. 7, pp. 133190–133202, 2019.
> >
> > [3] https://www.apple.com/uk/healthcare/products-platform/
> >
> > [4] C. Guo, G. Pleiss, Y. Sun, KQ. Weinberger. On calibration of modern neural networks. arXiv preprint arXiv:1706.04599. 2017 Jun 14.
> >
> > [5] Dai Z, Low BK, Jaillet P. Federated Bayesian optimization via Thompson sampling. Advances in Neural Information Processing Systems. 2020;33.
> >
> > [6] L. Corinzia, JM. Buhmann. Variational federated multi-task learning. arXiv preprint arXiv:1906.06268. 2019 Jun 14.

---

### Official Review · AnonReviewer2 · 2020-10-28
**An interesting paper with potential practical advantages over existing works but demonstration of these advantages against some existing works seem to be missing.**

**Rating:** 6
**Confidence:** 4

**Review:**

PAPER SUMMARY

This paper introduces a new approach to probabilistic federated learning, which builds on the previous PVI work of (Bui, 2018).

The proposed approach follows the same recipe in PVI where local agents learn their own model posteriors from private data, and communicate their posterior representations to a server, which aggregate local posterior representations into a universal representation. Local agents then download the aggregated posterior and offset it with their current posterior. The offsetted posterior is in turn used as the new local prior to re-run the corresponding local posterior approximation (via a generalized form of variational inference). New local posterior estimates are subsequently communicated to the server and so on.

However, unlike PVI, the proposed method aims to replace the parametric representation of posterior with a non-parametric particle representation developed by the prior SVGD work of (Liu & Wang, 2016). This necessitate the development of a distributed particle aggregation algorithm in Section 4, which is the key contribution of this work. This development is also motivated by two practical desiderata of federated learning: (a) a good trade-off between communication load (per iteration) and no. of communication iterations; and (b) well-calibrated predictions that are more trustworthy.

Following the above summary, I will give my opinions regarding several aspects of this paper below.

NOVELTY & SIGNIFICANCE

On the high level of idea, this paper presents an interesting perspective on a practical federated learning system: communication trade-off & trustworthy prediction. These are definitely important problems in the direction of making federated learning more efficient and robust. This is the novel angle that I like about this paper.

Its technical development, on the other hand, is leaning a bit more on the incremental side as the entire system is pretty much the same as that of PVI with the exception that a new particle representation is considered instead of PVI's parametric representation (in the statistical form of an exponential distribution).

A common pattern here is that both representations allow universal posterior information to factorize additively across local devices (in the respective forms of local posterior representation). In both cases, this leads to a variant of a distributed sum problem where each local party has some running estimate of some piece of local information & the goal is to communicate asynchronously so that each can refine its local estimate and eventually, recover the correct sum of information.

In the case of SVGD, however, the exact local update would require buffering all previous particle representations (i.e., past estimates) to date so that the downloaded posterior can be accurately offsetted to act as a prior for the local model (i.e. independent of local data). This necessitates the development of a distillation scheme in Section 4.2 which is, to me, the key technical contribution here. In addition, the theoretical analysis on the U-DVSGD's per-iteration decrease for the KL divergence is also an interesting contribution.

On this note, it seems the authors have deferred the demonstration of how well the KDE distillation approximate the original particle representation to various places in the appendix. Perhaps putting some of those back into the main text would be better (if space permits).

On the practical aspect of this paper (i.e. communication load & trustworthy prediction), while the demonstration is sufficient against point-estimate method such as FedAvg and DSGLD, there is no comparison against other non-parametric probabilistic methods such as PVI (Bui, 2018) and/or PNFM (Yurochkin, 2019). Given that the difference between PVI and DSVGD is a matter of posterior representation, comparison against PVI is probably necessary to showcase that the particle representation yields better calibrated predictions.

Also, the probabilistic non-parametric federated learning work of (Yurochkin, 2019) also allows multiple rounds of communication (although it can also be used as a one-shot model fusion of pre-trained local models) so it would be good to also compare both the communication load & the prediction caliberation against this work.

TECHNICAL SOUNDNESS

I have made high-level check of the derivations and have not found any technical issues.

CLARITY

The paper is very well-written, especially the part that summarizes the background on SVGD and PVI.

REVIEW SUMMARY

In short, this paper presents an interesting perspective on non-parametric probabilistic federated learning via particle representation of posterior. The technical development is sufficiently novel with demonstrated practical advantages against FedAvg and DSGLD. These practical advantages however were not demonstrated against existing probabilistic non-parametric federated learning works such as PVI & PNFM -- this is perhaps strange given that DVSGD builds on PVI and is mostly different only in terms of posterior representation.

--- post-rebuttal feedback ---

The authors have addressed most of my concerns. My rating for this paper therefore remains on the positive side.

---

> ### Author Response · Authors · 2020-11-16
> **Reply to reviewer**
>
> We would like first to thank the reviewer for the detailed assessment and excellent summary of our work and contributions, in addition for the positive evaluation. We address your two main concerns below.
>
>
> $\bullet$ Adding back from the Appendix the demonstration of how well the KDE distillation works
>
> Thank you. We agree that it is important to have a comparison of U-DSVGD, DSVGD and PVI. This has been done extensively, but only in the Appendix. As per you request, we have added Fig. 3 to the main text highlighting this comparison.
>
> $\bullet$ Demonstration against previous non-parametric work is missing
>
> We would like to first note that PVI is a parametric scheme as it requires the specification of a family of distributions to which the approximate parametric variational distribution $q(\theta;\eta)$ belongs. In contrast, DSGLD is a non-parametric technique as it is based on Monte Carlo (MC) sampling. We have tried to make this clearer in the text by defining a general distributed variational inference framework in Sec. 3, then its parametric implementation using PVI in Sec. 4 and finally our contributions: U-DSVGD and DSVGD in Sec. 5.
>
> In the submission, including the Appendix, we have used nine different benchmarks. These include a combination of centralized/decentralized and frequentist, variational Inference, and MC techniques. All the benchmarks used are summarized in Table 1 of the Appendix. We think that this benchmarking is extensive, and that we have covered comprehensively different classes of techniques.
>
> Finally, it is important to note that the goal of PFNM is aligned with frequentist learning rather than Bayesian learning: While Bayesian federated learning techniques, such as PVI and DSVGD, aim at obtaining the global posterior distribution in the model parameter space, the goal of PFNM is to produce a single model parameter vector. PFNM uses Bayesian non-parametrics to derive a probabilistic way of combining the model parameters of different neural networks in order to learn a more expressive global model. As such, the underlying learning framework is frequentist and not Bayesian.
>
> In summary, we believe that, while generally interesting, a comparison with PFNM is not required since it does not address the problem of Bayesian federated learning. That said, due to its relevance, we have cited and briefly reviewed PFNM in the related works section.

---

> > ### Comment · AnonReviewer2 · 2020-11-23
> > **Re: comparison with non-parametric method**
> >
> > Thank you for your response.
> >
> > I agree with you that the final outcome of PFNM is a point-based solution so perhaps comparing with PFNM does not fit perfectly within the scope of Bayesian federated learning.
> >
> > Nonetheless, such comparison could potentially show the advantage of being fully Bayesian so that might still be something that add extra values to the paper. But given your clarification above, this is more of a minor point so I leave it as a constructive suggestion to the authors. Not including it at this stage will not change my positive opinion of this paper.

---

> > > ### Author Response · Authors · 2020-11-24
> > > **reply  to reviewer**
> > >
> > > Thanks for you reply and for the positive feedback.

---

### Official Review · AnonReviewer3 · 2020-10-28
**Seem less correlated to federated learning**

**Rating:** 5
**Confidence:** 4

**Review:**

This paper proposes distributed SVGD, which maintains N particles both on the server and on the client. The communication between the server and the client is conducted by uploading/downloading these N particles. The learning of local client is formulated as inferring corresponding tilted distribution. Experiments are conducted on synthetic Gaussian 1D mixture, Bayesian logistic regression on Covertype and Twonorm dataset and Bayesian NN on the UCI dataset.

The idea of federated Bayesian learning is important and well motivated. However, using DSVGD for this purpose is not well supported in the paper. Non-iid data on each client, communication cost bottleneck and limited computational resources (either storage or computation) are three characteristics of federated learning. However, this paper pays little attention to them. The performance of SVGD relies on N (Liu, 2017) and more particles are needed for higher dimensional problems (e.g., NN). This property is unsuitable for federated learning. Since DSVGD needs to transfer N particles between the server and the client in each round, and  it needs to store and compute N particle in each client. The former increases the communication cost and the latter increases the burden of client. Besides, in the experiments, training dataset is randomly split into partition of equal size among the K agents, which follows the iid setting. Non-iid dataset partition is needed to evaluate thoroughly the performance of DSVGD for federated learning.

Some minor points:
1. Sec 3 SVGD: original SVGD use the particles directly as an approximation instead of using KDE
2. Eq. (14): the notation of $q^{(j)}$ and $q^{(i)}$ is confusing

---

> ### Author Response · Authors · 2020-11-16
> **Reply to reviewer (1/2)**
>
> We greatly appreciate the time you took to give specific comments of our work.
>
> To start with, it is important to reiterate that the paper contains not only plots of accuracy for  regression and classification, but also results concerning calibration, which highlight the trustworthiness of the proposed solution. Even though important, this type of studies seems to have been generally overlooked in the literature on federated learning, including classical frequentist learning algorithms like FedAvg [1] and FedProx [2] and their variants [3]. As we argue in our work, these frequentist solutions tend to perform poorly in terms of calibration. A specific example is shown in Fig. 8 of the main text. This is one of the key motivations for investigating Bayesian learning in a federated setting. We have revised Sec. 1 to clarify the motivations and limitations of this study, and you can find further details on this below where we address your concerns.
>
> *$\bullet$ Using DSVGD for Bayesian Learning is not well supported*
>
> The reviewer pointed out that the idea of using Bayesian learning is well motivated but using DSVGD for that purpose is not. We first note that, in Bayesian learning, two classes of practical implementations can be used, namely variational inference [Chapter 10, 4] and Monte Carlo sampling methods [Chapter 11, 4]. We specify below the key limitations of each class that motivate the introduction of DSVGD.
>
> •	Variational inference (VI): VI confines the variational posterior $q(\theta)$ to belong to a pre-specified set of parametric distributions -- typically from the exponential family of distributions [5] -- and parameters are updated via gradient descent methods. Naturally, this can cause a significant bias when the target posterior distribution cannot be well approximated by a distribution of that family. A typical example is in Fig. 10, Sec. B.1, where the target posterior is multimodal, while the chosen variational family is unimodal. When insufficient information is available on the shape of the posterior, e.g., on the number of modes, VI may not provide satisfactory performance. This is a key motivation for the introduction of non-parametric methods [Sec. 1, 8].
>
> •	Monte Carlo (MC) sampling: MC sampling approximates the target posterior by drawing approximate samples from the posterior, typically from a Markov chain whose transition probability matrix tends to the true posterior upon convergence. MC sampling has been adapted for use in federated settings, notably by DSGLD [6]. MC sampling is well known to be slow to converge. This is particularly problematic in federated settings, since a slow convergence implies a larger number of required communication rounds. This calls for the use of a non-parametric techniques that can converge more quickly than MC sampling.
> Using DSVGD, we have addressed simultaneously the limitations of both of the above classes as follows:
>
> •	Addressing VI limitations: DSVGD does not confine the variational posterior $q(\theta)$ to have a specific parametric form. By taking a non-parametric approach, it allows to approximate with high accuracy any target distribution.
>
> •	Addressing MC sampling limitations: DSVGD updates particles in a deterministic fashion, following a steepest descent direction in the KL divergence. As demonstrated in the large literature on SVGD, but not yet in the context of federated learning, this leads to much faster convergence speed than MC methods in terms of number of iterations, and hence number of communication rounds in federated learning which we extensively demonstrate throughout our experimental results.
> We have completely revised Sec. 1 to clarify these points in addition to modifying a part of Sec. 5.
>
> On a related note, due to the parametric constraint in VI methods, the fixed-point optimality property in Theorem 1 added to the revised version, does not apply for PVI, but it applied for DSVGD with a sufficiently large number of particles (in practice, a small number of particles was found to exhibit state-of-the-art performance as shown in our experiments and in [8]).

---

> > ### Author Response · Authors · 2020-11-16
> > **Reply to reviewer (2/2)**
> >
> > $\bullet$ Little attention is paid to the three characteristics of federated learning & Performance relying on the number of particles N is unsuitable for federated learning
> >
> >
> > As we point out in the revised Sec. 1 of the paper, federated learning applies at different scales, from industrial data silos to mobile devices, and each setting yields different types of challenges [9]. We are specifically interested in the small-scale federated learning regime consisting of mobile devices, each having a limited data set and running a small to average sized model due to memory constraints. Consider for instance the application of classifying activity – such as jogging, walking, and writing – based on data from an accelerometer [10], or the implementation of a health care monitor based on data from smart-watch ECG data [11]. A model with a limited number of parameters – say a few hundreds of weights – would be sufficient for the problem at hand, and each device would have a small number of data points.
> > In this context, non-iid data, communication cost bottleneck, and limited computational resources, are all relevant, but one also need to consider the following additional challenges:
> >
> > •	Trustworthiness: Frequentist approaches are badly calibrated [7], as shown in Fig. 8 of our paper, which makes their use in critical applications domain such as healthcare monitoring, ill advised. This problem has been largely overlooked in the federated learning community;
> >
> > •	Number of communication rounds: When models are small, the payload per communication round may not be the main contributor to the overall latency of the training process. In contrast, accommodating many communication rounds requiring arbitrating channel access among multiple devices and may yield slow clock time convergence.
> >
> > We believe that the proposed approach successfully addresses both these challenges for the first time. In the paper, we show that it provides well-calibrated decisions and it can vastly improve the convergence speed in terms of communication rounds. It is true that each communication round requires a larger communication load since multiple particles need to be exchanged. While this is certainly a limitation for larger-scales implementations of federated learning, we argue that, for smaller-scale deployments, the trade-off between per-iteration communication load and number of communication rounds is well worth exploring and practically relevant. To see this, as an example, consider the accuracy using the covertype dataset in Fig. 3 on the right-hand side. The reach an accuracy of 70% using DSVGD with $N=6$ particles, only $5$ communication rounds are required. This means that the total communication load in terms of particles exchanged for DSVGD is $5\times6 = 30$, while, for FedAvg, around $100$ rounds are needed to reach the same accuracy, thus making the total communication load equal to $100$, which is much more than the communication load of DSVGD. We have further clarified this in Sec. 5.
> >
> > That being said, we agree that we did not properly account for the aspect of “non-iidness”, which is an important requirement. In the revised paper, we have added a new plot in Fig. 5 to show that Bayesian federated learning, and specifically DSVGD, is more robust to the presence of non-iid data sets as compared to frequentist federated learning techniques. Fig. 5 specifically shows the test log-likelihood for the covertype data set. It is seen that, as the split of classes among the devices becomes more skewed, the performance of DSVGD is hardly affected, while FedAvg is significantly impaired. The result hinges on the fact that Bayesian learning provides a predictive distribution that is a more accurate estimate of the ground-truth posterior distribution. This is true irrespective of the level of “non-iidness”: Bayesian learning can account in a principled way for all competing “explanations” provided by different devices of the data set. This is in contrast to FedAvg, whose reliance on a point estimate of the parameters yields an overconfident predictive distribution that cannot properly account for the diversity of predictions provided by different devices.
> >
> >  $\bullet$ Minor Points
> >
> > 1.	Yes, SVGD uses the original particles throughout the algorithm and a KDE is only needed at the end to approximate the distribution of the particles to be used (please see, for e.g., Fig 1 in [12]).
> >
> > 2.	Sorry for this confusion, $q^{(j)}$ simply denotes the approximate posterior during the communication rounds $j \leq i-1$ when the agent at hand is scheduled.

---

> > > ### Author Response · Authors · 2020-11-16
> > > **References used in replies**
> > >
> > > [1] B. McMahan, E. Moore, D. Ramage, S. Hampson, BA Arcas. Communication-efficient learning of deep networks from decentralized data. InArtificial Intelligence and Statistics 2017 Apr 10 (pp. 1273-1282). PMLR.
> > >
> > > [2] T. Li, AK. Sahu, M. Zaheer, M. Sanjabi, A. Talwalkar, V. Smith. Federated optimization in heterogeneous networks. Proceedings of Machine Learning and Systems. 2020 Mar 15;2:429-50.
> > >
> > > [3] R. Pathak, MJ. Wainwright. FedSplit: An algorithmic framework for fast federated optimization. arXiv preprint arXiv:2005.05238. 2020 May 11.
> > >
> > > [4] CM. Bishop. Pattern recognition and machine learning. springer; 2006.
> > >
> > > [5] M. Wainwright & M. Jordan. Graphical models, exponential families, and variational inference. Now Publishers Inc; 2008.
> > >
> > > [6] S. Ahn, B. Shahbaba, M. Welling. Distributed stochastic gradient MCMC. In International conference on machine learning 2014 Jan 27 (pp. 1044-1052).
> > >
> > > [7] C. Guo, G. Pleiss, Y. Sun, KQ. Weinberger. On calibration of modern neural networks. arXiv preprint arXiv:1706.04599. 2017 Jun 14.
> > >
> > > [8] Q. Liu, D. Wang. Stein variational gradient descent: A general purpose bayesian inference algorithm. In Advances in neural information processing systems 2016 (pp. 2378-2386).
> > >
> > > [9] Kairouz P, McMahan HB, Avent B, Bellet A, Bennis M, Bhagoji AN, Bonawitz K, Charles Z, Cormode G, Cummings R, d'Oliveira RG. Advances and open problems in federated learning. arXiv preprint arXiv:1912.04977. 2019 Dec 10.
> > >
> > > [10] Mary M Weiss, Kenichi Yoneda, and Thaier Hayajneh, “Smartphoneand smartwatch-based biometrics using activities of daily living,” IEEE Access, vol. 7, pp. 133190–133202, 2019.
> > >
> > > [11] https://www.apple.com/uk/healthcare/products-platform/
> > >
> > > [12] Liu Q, Wang D. Stein variational gradient descent: A general purpose bayesian inference algorithm. InAdvances in neural information processing systems 2016 (pp. 2378-2386).
> > >
> > > [13] Bui TD, Nguyen CV, Swaroop S, Turner RE. Partitioned variational inference: A unified framework encompassing federated and continual learning. arXiv preprint arXiv:1811.11206. 2018 Nov 27.

---

### Official Review · AnonReviewer1 · 2020-10-28
**Official Blind Review #1**

**Rating:** 5
**Confidence:** 3

**Review:**

This paper proposes a Bayesian optimization algorithm in the context of federated learning. The whole framework is built on top of generalized Bayesian learning. To overcome the locality of clients' distributions, the authors propose their solution as an integration of Partitioned Variational Inference (PVI) and Stein Variational Gradient Descent (SVGD). Numerical experiments have been conducted on a synthetic dataset and some standard benchmark datasets, and evaluated on both regression and classification tasks.

The problem of federated Bayesian learning is important, especially in the time when communication and data privacy arise public attention. The algorithm proposed is interesting and has shown benefits in the experiments.  I have a few comments as follows:

1. The organization of the paper needs to be revised. Compared with PVI and SVGD, the proposed algorithm seems too long and too complicated, and the algorithm is not shown in the body but in Appendix. I think it might be better to decompose the proposed algorithms into several components, e.g., the server part and the agent part, and packing important updating steps as a procedure.

2. The motivation is not clearly stated in this paper. The comparison with PVI should be extended with more details. For example, it seems to me that the authors use a different way to optimize the local free energy functional, compared with PVI.  If my understanding is wrong please correct me. But if so, what's the motivation of doing so? Why we should consider SVGD instead of natural gradient for this subproblem?

3. I would also like to see more theoretical understanding of the proposed algorithm. For example, the convergence analysis for (U-)SVGD, even for the most simplified case, and compares with the existing work. Also the density evolution and the fixed-points analysis are also good to include. I do see some analysis in the appendix, but I think some results are better to be present in the main text.

4. The experiments are conducted on synthetic or small datasets. I think the authors should include experiments on larger datasets and/or more complicated models.

5. I'm not sure if "Global iteration index" is a common term but in federated learning (e.g., the FedAvg paper), it's usually called "communication rounds".

6. A quick question: the proposed algorithm and PVI only selects ONE agent per communication round. What if more agents can be selected in a single round, like FedAvg does?

---

> ### Author Response · Authors · 2020-11-16
> **Reply to reviewer (1/2)**
>
> We thank the reviewer for the detailed assessment and useful suggestions that have helped to improve our work. We address the key issues below.
>
> **1.** Thank you for your suggestion. We have made this change to the current version of the paper. Given the reduced size of the table, it has now been included in the main text in Algorithm 1.
>
> **2.** The reviewer is correct: we indeed rely on SVGD to minimize the local free energy functional. The main motivation for doing this is that SVGD provides substantial improvements over classical gradient based techniques by enabling the use of non-parametric representations of the variational posterior $q(\theta)$. This is in line with the original motivation for using SVGD in lieu of conventional parametric variational inference [Sec. 1,4].
> To elaborate on this point, minimizing the local free energy via any gradient based method would require constraining the variational distribution $q(\theta)$ to a specific parametric family of distributions -- typically selected from the exponential family [1] -- and then updating the parameters of $q(\theta)$ via gradient/natural gradient descent while being constrained within this specific family. This is known to yield poor performance if the chosen family of variational distributions cannot capture the target true posterior, yielding a biased estimate, and the fixed-point property in Theorem 1 (see reply for your next question) does not hold. An example is the case where the target posterior is multimodal, while the chosen family is unimodal. In contrast, by following a non-parametric approach, SVGD approximates the target posterior via a set of particles that can be easily adjusted to capture any target posterior without being constrained to having a specific shape, e.g., a given pre-specified number of modes. Please see Fig 10 in Sec. B.1 of the Appendix for an illustration and a graphical comparison between SVGD/DSVGD and PVI.
> We have revised Sec. 1 in order to clarify the motivation of the work. Furthermore, we have introduced a new section (Sec. 3) that defines a more generic framework called DVI. Then, we discuss separately PVI as a constrained instance of DVI and further motivate DSVGD in Sec. 5.
>
> **3.** Thank you for this useful suggestion. We first note that, as PVI, the proposed algorithm is an Expectation Propagation (EP)-like method [3]. When no constraints are imposed on the form of the variational posterior $q(\theta)$ and the optimization of the local free energy is exact, the unique fixed points of the local free energy optimizations coincides with the optimal solution of the global free energy (see Theorem 1). This offers further understanding and theoretical motivation to our algorithm. We have added a new theorem highlighting this fixed-point property in Sec 3.
> When imposing a specific parametrization as done in PVI, this fixed-point property is not necessarily guaranteed. In contrast, with DSVGD, in the limit of a large number of particles, it is possible to approximate arbitrarily closely the unconstrained solution [10], and hence the fixed-point property mentioned above applies. We have made this clear in the revised paper.
> Regarding convergence, it is well understood that EP-like schemes, including PVI, are not theoretically guaranteed to converge. DSVGD inherits this limitation, although empirical evidence – for EP, PVI, and DSVGD – confirms excellent convergence properties. That said, we have been able to prove a guaranteed improvement result that quantifies the decrease of the global free energy at each local update step. The result, which we have now included in Sec. 5.1 of the main text, demonstrates that, by properly choosing the learning rate, it is in principle possible to ensure that the global free energy is non-increasing across the iterations. While this does not prove convergence to a stationary point, to the best of our knowledge, it provides the strongest type of convergence guarantees that is to be expected for this class of algorithms.
>
> **4.** Considering the Appendix, we have tested our algorithm with $16$ different datasets of different sizes. While on the one hand we have used some of the classical datasets that are typically used in the federated learning literature (for e.g., MNIST [6], F-MNIST and Vehicle [5]), it is true that we have also added results with much smaller datasets (for e.g., diabetes and breast cancer). This was done for the following reasons:
>
> •	We are interested in capturing federated learning scenarios with agents that may not have large data sets, neither individually nor collectively. An example is given by wireless sensors or IoT devices;
>
> •	Frequentist approaches such FedAvg [7] tend to perform poorly in the low data regime, both from an accuracy perspective (due to overfitting) and in terms of calibration. In this low-data regime, Bayesian methods such as DSVGD are a natural solution to investigate.
>
> We have clarified this scenario in Sec. 1.

---

> > ### Author Response · Authors · 2020-11-16
> > **Reply to reviewer (2/2)**
> >
> > **5.** Thank you for pointing this out. Indeed, in the federated learning literature the term “communication rounds” is more common than “global iterations”, which is typically used in the distributed Bayesian inference literature. We have changed this accordingly throughout the text and all the figures.
> >
> > **6.**  This is a very important point that we had left as a future work in the conclusion of the original submission. It is possible to schedule multiple agents in parallel, and we have investigated this direction in the revised paper. To elaborate, consider first the unconstrained optimization of the variational posterior, which we cover in Sec. 3 of the revised paper. In this case, scheduled agents $k \in \mathcal{K}^{(i)}$ in round $i$ share their local approximate likelihood’s parameter $\eta_{k}^{(i)}$, and then the server construct the global approximate posterior as
> >
> > $ q^{(i)}(\theta) = p_0(\theta) \prod_{k \in \mathcal{K}^{(i)}} t_{k}^{(i)}(\theta) \prod_{k^\prime \not\in \mathcal{K}^{(i)}} t_{k^\prime}^{(i)}(\theta)$, (1)
> >
> > where $t_{k^\prime}^{(i)} (\theta) = t_{k^\prime}^{(i-1)} (\theta) $ and $t_{k}^{(i)} (\theta) $ are updated as in equation (7) of the revised paper.
> > At a theoretical level, the performance of the resulting scheme benefits from the same fixed-point property of the scheme that schedules a single device in each communication round. In particular, as we detail in Theorem 1 of the revised paper, the unique fixed point of this procedure is the global optimal posterior.
> > Based on this theoretical insight, in Sec A.5 of the Appendix of the revised paper, we introduce a parallel extension of DSVGD, termed Parallel-DSVGD (P-DSVGD). In P-DSVGD, during the $i$-th communication round, scheduled agents report the local particles $ \theta_{k,n}^{(i)} $ encoding the approximate likelihoods $t_{k}^{(i)}(\theta)$, for $k \in \mathcal{K}^{(i)}$, to the server. Then, to approximate $q^{(i)}(\theta)$ as in (1), we propose the use of server side particles $\phi_{n}^{(i)} $. The latter are updated via SVGD to approximate (1). Then, the new set of scheduled agent $\mathcal{K}^{(i+1)}$ download the updated $\phi_{n}^{(i+1)} $ and carries out the same steps as in DSVGD.
> > Sec. A.5 of the Appendix reports also corresponding experimental results on several datasets.
> >
> > Thank you.
> >
> > **References**
> >
> > [1] M. Wainwright & M. Jordan. Graphical models, exponential families, and variational inference. Now Publishers Inc; 2008.
> >
> > [2] A. Korba, A. Salim, M. Arbel, G. Luise, A. Gretton. A Non-Asymptotic Analysis for Stein Variational Gradient Descent. arXiv preprint arXiv:2006.09797. 2020 Jun 17.
> >
> > [3] TP. Minka. Expectation propagation for approximate Bayesian inference. arXiv preprint arXiv:1301.2294. 2013 Jan 10.
> >
> > [4] Q. Liu, D. Wang. Stein variational gradient descent: A general purpose bayesian inference algorithm. In Advances in neural information processing systems 2016 (pp. 2378-2386).
> >
> > [5] T. Li, M. Sanjabi, A. Beirami, V. Smith. Fair resource allocation in federated learning. arXiv preprint arXiv:1905.10497. 2019 May 25.
> >
> > [6] M. Yurochkin, M. Agarwal, S. Ghosh, K. Greenewald, TN. Hoang, Y. Khazaeni. Bayesian nonparametric federated learning of neural networks. arXiv preprint arXiv:1905.12022. 2019 May 28.
> >
> > [7] B. McMahan, E. Moore, D. Ramage, S. Hampson, BA Arcas. Communication-efficient learning of deep networks from decentralized data. InArtificial Intelligence and Statistics 2017 Apr 10 (pp. 1273-1282). PMLR.
> >
> > [8] H. Chen, W. Chao. FedBe: Making Bayesian Model Ensemble Applicable to Federated Learning
> >
> > [9] Bui TD, Nguyen CV, Swaroop S, Turner RE. Partitioned variational inference: A unified framework encompassing federated and continual learning. arXiv preprint arXiv:1811.11206. 2018 Nov 27.
> >
> > [10] Q. Liu. Stein variational gradient descent as gradient flow. In Advances in neural information processing systems 2017 (pp. 3115-3123).

---

### Author Response · Authors · 2020-11-16
**Revised paper**

Dear reviewers,

We would like to thank all of you for the detailed comments and feedback we have received on our work. By taking them into consideration, we believe that we have significantly improved the manuscript. We have now added new theoretical, algorithmic and experimental results. We have proposed a parallelizable version of our algorithm, and completely re-worked some of the sections of the paper in order to justify and clarify the theory and intuition behind the proposed solutions.

For your convenience, all modifications in the revised version of the paper are made using the blue color.

We summarize below our main contributions, the main common concerns, and how we addressed them.

**Main Contributions**

Existing works in the federated learning literature mainly focus on frequentist approaches that target accuracy as the performance criterion of interest by considering high-capacity models that require model compression to reduce the communication load. In contrast, this paper concentrates on settings consisting of mobile or embedded devices, each having a limited data set and running a small-sized model, for which trustworthiness and number of communication rounds are essential performance criteria. An example is given by personal health monitors based on smart watch ECG signals. Bayesian federated learning is adopted as a principled approach to address these requirements, which are not met by current frequentist solutions. Moving beyond existing parametric and Monte Carlo federated Bayesian learning protocols, we have introduced a non-parametric Bayesian federated learning algorithm that is not constrained by the bias of parametric variational approximations and exhibits fast convergence in terms of number of communication rounds. Theoretical fixed-point guarantees are provided, along with a bound on the per-iteration decrease of the global free energy objective. Furthermore, the proposed algorithm is seen to exhibit strong performance on non-iid datasets due to its Bayesian nature.

**Main common concerns**

1.	The presented approach has limited practical use because of the current restrictions it imposes (i.e. update one agent at a time)

2.	The presented approach increases the communication load per round which seems against federated learning main goals.

**Summary of Proposed solutions**

1.	We have addressed this limitation by proposing Parallel-DSVGD (P-DSVGD), a direct extension of DSVGD that allows multiple devices to be scheduled per communication round. P-DSVGD preserves the non-parametric property of DSVGD and does not require any extra computational effort at the devices. Corresponding theoretical and experimental results were added to the paper.

2.	As mentioned, this paper focuses on settings consisting of mobile or embedded devices, each having a limited data set and running a small-sized model, in which low-latency and trustworthiness are essential performance criteria. In these settings, the per-iteration communication load amounts to the transmission of a limited number of weights, and the latency of the overall training procedure is dominated by the number of communication rounds. That said, even though DSVGD requires a larger communication load per communication round, the total communication load across all communication rounds can in practice be much lower than frequentist techniques, due to the faster convergence of DSVGD. This is discussed thoroughly in the revised introduction section and verified via numerical results.

Please see below for individual detailed responses for all your concerns. Please don’t hesitate to get in touch for any further suggestions or comments.

Thank you.

---

### Decision · Program_Chairs · 2021-01-07
**Final Decision**

**Decision:**

Reject

**Comment:**

This work presents a distributed SVGD (DSVGD) algorithm as a new non-parametric Bayesian framework for federated learning. The reviewers concerned with the practical advantages of the proposed method, including the communication cost and the constraint of updating one agent per time. The authors rebuttal helped addressing some of the concerns, including proposing a new Parallel-DSVGD algorithm. This is very much appreciated. However, given the significant modification needed over the original version, we think it is better for the authors to further improve the work and submit to the next conference.